# Genetic architecture and lifetime dynamics of inbreeding depression in a wild mammal

M. A. Stoffel [1✉], S. E. Johnston [1], J. G. Pilkington[1] & J. M. Pemberton[1]

Inbreeding depression is ubiquitous, but we still know little about its genetic architecture and precise effects in wild populations. Here, we combine long-term life-history data with 417 K imputed SNP genotypes for 5952 wild Soay sheep to explore inbreeding depression on a key fitness component, annual survival. Inbreeding manifests in long runs of homozygosity (ROH), which make up nearly half of the genome in the most inbred individuals. The ROH landscape varies widely across the genome, with islands where up to 87% and deserts where only 4% of individuals have ROH. The fitness consequences of inbreeding are severe; a 10% increase in individual inbreeding $F_{ROH}$ is associated with a 60% reduction in the odds of survival in lambs, though inbreeding depression decreases with age. Finally, a genome-wide association scan on ROH shows that many loci with small effects and five loci with larger effects contribute to inbreeding depression in survival.

[1] Institute of Evolutionary Biology, School of Biological Sciences, University of Edinburgh, Edinburgh, UK. ✉email: martin.stoffel@ed.ac.uk

nbreeding depression, the reduced fitness of offspring from related parents, has been a core theme in evolutionary and conservation biology since Darwin[1]. The detrimental effects of inbreeding on a broad range of traits, individual fitness and population viability have now been recognised across the animal and plant kingdoms[1–9]. With the ongoing decline of animal populations[10] and global habitat fragmentation[11], rates of inbreeding are likely to accelerate, and so it is increasingly important to have a detailed understanding of its genetic causes and fitness consequences to inform conservation strategies. However, we still know very little about some of the most fundamental features of inbreeding depression in wild populations[7,12], such as its precise strength, how it varies across life-history stages[13–15], and if it is driven by loci with weak and/or strong deleterious effects on fitness. As genomic data proliferates for wild populations, it is increasingly possible to quantify the distribution of effect sizes at loci underpinning inbreeding depression. By determining this genetic architecture, we can improve our understanding of the relationship between inbreeding, purging and genetic rescue[7,16], with important implications for the persistence and conservation of small populations[17–19].

Inbreeding decreases fitness because it increases the fraction of the genome which is homozygous and identical-by-descent (IBD). This unmasks the effects of (partially-) recessive deleterious alleles or in rarer cases may decrease fitness at loci with heterozygote advantage[4,20]. While the probability of IBD at a genetic locus was traditionally estimated as the expected inbreeding coefficient based on a pedigree[21,22], modern genomic approaches enable us to gain a much more detailed picture. Genome-wide markers or whole-genome sequences are now helping to unravel the genomic mosaic of homo- and heterozygosity and, unlike pedigree-based approaches, capture individual variation in homozygosity due to the stochastic effects of Mendelian segregation and recombination[12,23]. This makes it possible to quantify realised rather than expected individual inbreeding, and to measure IBD precisely along the genome[24,25].

An intuitive and powerful way of measuring IBD is through runs of homozygosity (ROH), which are long stretches of homozygous genotypes[26]. An ROH arises when two IBD haplotypes come together in an individual, which happens more frequently with increasing parental relatedness. The frequency and length of ROH in a population vary along the genome due to factors such as recombination, gene density, genetic drift and linkage disequilibrium[27–31] and extreme regions can potentially pinpoint loci under natural selection[24,31]. Regions with high ROH density, known as 'ROH islands'[32], have low genetic diversity and high homozygosity and have been linked to loci under positive selection in humans[31]. Regions where ROH are rare in the population, known as 'ROH deserts', could be due to loci under balancing selection or loci harbouring strongly deleterious mutations under purifying selection[12,24,33–35]. Moreover, the abundance of ROH also varies among ROH length classes, which are shaped by the effective population size ($N_e$)[30,31,36,37] and can vary in their deleterious allele load[38–41]. Longer ROH are a consequence of closer inbreeding and their relative abundance is indicative of recent $N_e$. This is because their underlying IBD haplotypes have a most recent common ancestor (MRCA) in the recent past with fewer generations for recombination to break them up. In contrast, shorter ROH are derived from more distant ancestors and their relative abundance in the population reflects $N_e$ further back in time[42].

Individual inbreeding can be measured as the proportion of the autosomal genome in ROH ($F_{ROH}$), which is an estimate of realised individual inbreeding $F$[43]. $F_{ROH}$ has helped to uncover inbreeding depression in a wide range of traits in humans and farm animals[5,29,44] and is often preferable to other SNP-based

inbreeding estimators in terms of precision and bias[5,45–47]. While $F_{ROH}$ condenses the information about an individual's IBD into a single number, quantifying the genomic locations of ROH across individuals makes it possible to identify the loci contributing to inbreeding depression and estimate their effect sizes[12]. As mapping inbreeding depression in fitness and complex traits requires large samples in addition to dense genomic data[37], the genetic architecture of inbreeding depression has mostly been studied in humans and livestock[48,49]. However, individual fitness will be different under natural conditions and consequently, there is a need to study inbreeding depression in wild populations to understand its genetic basis in an evolutionary and ecological context. To date, only a handful of studies have estimated inbreeding depression using genomic data in the wild[13,14,50–53]. While these studies show that inbreeding depression in wild populations is more prevalent and more severe than previously thought, all of them used genome-wide inbreeding coefficients and did not explore the underlying genetic basis of depression.

The Soay sheep of St. Kilda provides an exceptional opportunity for a detailed genomic study of inbreeding depression. The Soay sheep is a primitive breed that was brought to the Scottish St. Kilda archipelago around 4000 years ago[54], and has survived on the island of Soay ever since. Although the Soay sheep have been largely unmanaged on Soay, there is written and genomic evidence of an admixture event with the now-extinct Dunface breed ~150 years or 32 generations ago[55]. In 1932, 107 Soay sheep were transferred to the neighbouring island of Hirta where they are unmanaged. On Hirta the population increased and nowadays fluctuates between 600 and 2200 individuals. A part of the population in Village Bay became the subject of a long-term individual-based study, and detailed life history, pedigree and genotype data have been collected for most individuals born since 1985[54].

Here, we combine annual survival data for 5952 free-living Soay sheep over a 35 year period with 417 K partially imputed genome-wide SNP markers to estimate the precise effects and genetic basis of inbreeding depression. First, we quantify the genomic consequences of inbreeding through patterns of ROH among individuals and across the genome. We then calculate individual genomic inbreeding coefficients $F_{ROH}$ to model inbreeding depression in annual survival and estimate its strength and dynamics across the lifetime. Finally, we explore the genetic architecture of inbreeding depression using a mixed-model-based genome-wide association scan on ROH to shed light on whether depression is caused by many loci with small effects, few loci with large effects or a mixture of both.

## Results

**Genotyping and imputation.** All study individuals have been genotyped on the Illumina Ovine SNP50 BeadChip assaying 51,135 SNPs. In addition, 189 individuals have been genotyped on the Ovine Infinium High-Density chip containing 606,066 SNPs. To increase the genomic resolution for our analyses, we combined autosomal genotypes from both SNP chips with pedigree information to impute missing SNPs in individuals genotyped at lower marker density using AlphaImpute[56]. Cross-validation showed that imputation was successful, with a median of 99.3% correctly imputed genotypes per individual (Supplementary Table 1). Moreover, the inferred inbreeding coefficients $F_{ROH}$ were very similar when comparing individuals genotyped on the high-density chip (median $F_{ROH} = 0.239$) and individuals with imputed SNPs (median $F_{ROH} = 0.241$), indicating no obvious bias in the abundance of inferred ROH based on imputed data (Supplementary Fig. 1). After quality control, the genomic dataset contained 417,373 polymorphic and autosomal SNPs with

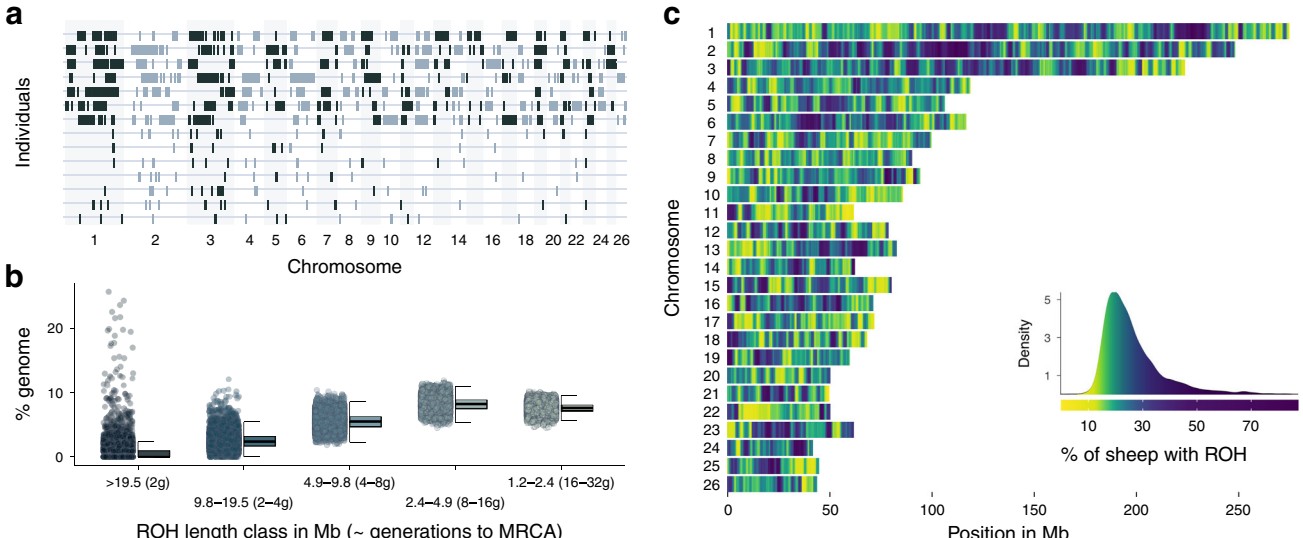

**Fig. 1 Runs of homozygosity (ROH) variation among individuals and across the genome. a** ROH longer than 5 Mb in the seven individuals with the highest inbreeding coefficients $F_{ROH}$ in the seven top rows and the seven individuals with the lowest $F_{ROH}$ in the seven bottom rows. **b** Distribution of ROH among different length classes. Each data point represents the proportion of ROH of a certain length class within an individual's autosomal genome. ROH length classes were categorised by their expected average physical length when the underlying haplotypes had a most recent common ancestor (MRCA) 2–32 generations (g) ago. **c** Genome-wide ROH density among all 5952 individuals in non-overlapping 500 Kb windows. The colour gradient has been scaled according to the ROH density, which is shown in the figure legend.

a mean minor allele frequency (MAF) of 23% (Supplementary Fig. 2) and a mean call rate of 99.5% across individuals.

**Patterns of inbreeding in the genome**. We first explored how inbreeding and long-term small population size (estimated $N_e$ = 194 [57])-shaped patterns of ROH in Soay sheep (Fig. 1). Individuals had a mean of 194 ROH (sd = 11.6) longer than 1.2 Mb, which on average made up 24% of the autosomal genome (i.e. mean $F_{ROH}$ = 0.24, range = 0.18–0.50, Supplementary Fig. 3). The mean individual inbreeding coefficient $F_{ROH}$ of sheep born in a given year remained constant over the course of the study period (Supplementary Fig. 4). Among individual variation in ROH length was high: the average ROH in the seven most inbred sheep was more than twice as long as ROH in the seven least inbred sheep (6.83 vs. 2.72 Mb, respectively; Fig. 1a) although the average number of ROH was similar (170 vs. 169, respectively).

The abundance of ROH in Soay sheep also varied considerably among ROH length classes (Fig. 1b). The largest fraction of IBD in the population consisted of ROH between 2.4 and 4.9 Mb originating around 8–16 generations ago, which made up 8.1% of an individual's genome on average. Long ROH > 19.5 Mb were found in 38.2% of individuals (Fig. 1b). However, long ROH made up on average only 0.6% of the genome of the least inbred individuals with pedigree inbreeding $F_{ped}$ < 0.1. In contrast, long ROH extended over 7% and 18% of the genome in inbred individuals with $F_{ped}$ > 0.1 and $F_{ped}$ > 0.2, respectively (Supplementary Fig. 5).

The frequency of ROH in the population varied widely across the genome (Fig. 1c). We scanned ROH in non-overlapping 500 Kb windows, and classified the 0.5% windows with the highest ROH density as ROH islands and the 0.5% windows with the lowest ROH density as ROH deserts [31]. The top ROH island on chromosome 1 (227–227.5 Mb) contained ROH in 87% of individuals, while only 4.4% of individuals had an ROH in the top ROH desert on chromosome 11 (58.5–59 Mb, see Supplementary Table 2 for a list of the top ROH deserts and islands).

**ROH density and recombination rate**. The wide variation in ROH density along the genome could be partially explained by recombination because regions with high recombination rate produce shorter ROH, and these are less likely to be detected by ROH calling algorithms [30]. Notably, recombination by itself does not impact the underlying true proportion of IBD, but only affects the ROH length distribution. Consequently, regions with high recombination could create putative ROH deserts without a change in the true levels of IBD, because short ROH are less likely to be called [30]. To evaluate how much variation in ROH density along the genome is due to recombination rate variation and how much of it is tracking the underlying levels of IBD, we constructed a linear mixed model with ROH density (proportion of individuals with ROH) measured in 500 Kb windows as response variable, window recombination rate in cM/Mb based on the Soay sheep linkage map [58] and window SNP heterozygosity as fixed effects as well as a chromosome identifier as a random effect.

Recombination rate and heterozygosity together explained 42% of the variation in ROH density (marginal $R^2$ = 0.42, 95% CI [0.40, 0.44], Supplementary Table 3A), with the majority of variation explained by heterozygosity (semi-partial $R^2$ = 0.38, 95% CI [0.36, 0.40], Fig. 2b) and only around 4% explained by recombination rate (semi-partial $R^2$ = 0.04, 95% CI [0.02, 0.07], Fig. 2a and Supplementary Fig. 6 for a chromosome-wise plot). The pattern is similar when re-running the model only on windows identified as ROH islands and deserts, where ROH density is largely explained by heterozygosity (semi-partial $R^2$ = 0.89, 95% CI [0.83, 0.94], Fig. 2d), with only a small proportion of the variation explained by recombination rate variation (semi-partial $R^2$ = 0.07, 95% CI [0.01, 0.12], Fig. 2c). Consequently, although recombination rate impacts ROH lengths and hence ROH detection probabilities, this accounts for only a small proportion of the variation in detected ROH density, which mostly reflects the underlying patterns of IBD along the genome.

Lastly, we explored how much variation in ROH density was explained by recombination when using different minimum ROH thresholds. We repeated the analysis with a dataset based on a minimum ROH length threshold of 0.4 Mb, and a second dataset

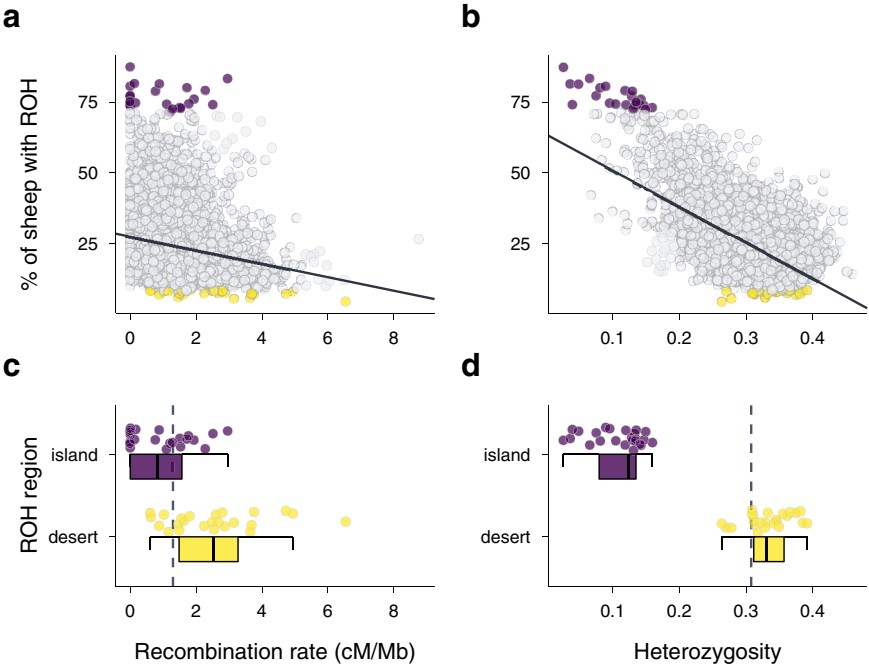

**Fig. 2 Correlates of ROH density variation across the genome.** Runs of homozygosity (ROH) density, recombination rate and heterozygosity were quantified in non-overlapping 500 Kb windows, with each point representing one window. The top 0.5% of windows with the highest and lowest ROH density in the population, termed ROH islands ($n = 24$) and deserts ($n = 24$), are coloured in purple and yellow respectively in all four plots. **a** Relationship between ROH density and recombination rate. **b** Relationship of ROH density and SNP heterozygosity. **c** Recombination rate within ROH islands and deserts. **d** Heterozygosity within ROH islands and deserts. Solid lines in a and b are linear regression lines and dashed lines in c and d are genome-wide means. Boxplots show the median as a centre line with the bounds of the box as 25th and 75th percentiles and upper and lower whiskers as largest and smallest value but no further than 1.5 * inter-quartile range from the hinge. Source data for this figure are also provided as a source data file.

with a minimum ROH length of 3 Mb (Supplementary Table 3B, C). Compared to the original dataset with a minimum ROH length of 1.2 Mb, recombination explained less variation in ROH density in the dataset including shorter ROH (semi-partial $R^2 = 0.01$, 95% CI [0.00, 0.04]) and more variation in the dataset consisting only of longer ROH (semi-partial $R^2 = 0.08$, 95% CI [0.06, 0.11]). Consequently, recombination rate variation has a larger impact on the detected abundance of longer ROH across the genome.

**Inbreeding depression in survival**. Survival is a key fitness component. In Soay sheep, more than half of all individuals die over their first winter, minimising their chances to reproduce (Supplementary Fig. 7). Sheep survival is assessed through routine mortality checks which are conducted throughout the year. Over 80% of sheep in the study area are found after their death[50], resulting in a total of 15,889 annual survival observations for 5952 sheep. The distribution of individual inbreeding coefficients $F_{ROH}$ in different age classes revealed that highly inbred individuals rarely survive their early years of life and never reach old ages (Fig. 3a). However, the strength of inbreeding depression appeared to decline at older ages (Fig. 3b). For example, in sheep older than four years, the proportion of survivors among the most inbred individuals was only marginally lower than among the least inbred individuals (Fig. 3b).

We modelled the strength of inbreeding depression across the lifetime using an animal model with a binomial error distribution and annual survival as a response variable. Overall, the effect of inbreeding on survival was strong: in lambs (age 0), a 10% increase in $F_{ROH}$ was associated with a 0.4 multiplicative change in the odds of survival (odds ratio, OR [95% credible interval, CI] = 0.40 [0.30, 0.53], Supplementary Table 4), or a 60% reduction (1−0.40) in the odds of survival. This translates into non-linear survival

differences on the probability scale. For example, a male non-twin Soay sheep lamb with an $F_{ROH}$ 10% above the mean had a 23% lower probability of surviving its first winter compared to an average lamb ($F_{ROH} = 0.34$ vs $F_{ROH} = 0.24$; Fig. 3c). Across the lifetime, the model estimates for the interactions between $F_{ROH}$ and the different life stages predicted a decrease in the strength of inbreeding depression in later life stages (Fig. 3c) with the largest predicted difference between early (age 1, 2) and late-life (age 5+, OR [95% CI] = 2.03 [1.08, 3.82], Supplementary Table 4).

We next estimated the inbreeding load in Soay sheep as the diploid number of lethal equivalents $2B$. Lethal equivalents are a concept rooted in population genetics, where one lethal equivalent is equal to a group of mutations which, if dispersed across individuals, would cause one death on average[59]. We followed suggestions by Nietlisbach et al.[60] and refitted the survival model with a Poisson distribution and logarithmic link function using a simplified model structure without interactions for better comparability across studies. This gave an estimate of $2B = 4.57$ (95% CI 2.61–6.55) lethal equivalents for Soay sheep annual survival.

**Genetic architecture of inbreeding depression**. To quantify the survival consequences of being IBD at each SNP location, we used a modified genome-wide association study (GWAS). Unlike in traditional GWAS where p-values of additive SNP effects are of interest, we analysed the effects of ROH status for both alleles at every SNP. Specifically, at a diallelic locus, ROH result either from two IBD haplotypes containing allele A or from two IBD haplotypes containing allele B. If strongly deleterious recessive alleles exist in the population, they could be associated with ROH based on allele A haplotypes or ROH based on allele B haplotypes. To test this, we constructed a binomial mixed model of annual survival for each SNP position. In each model, we fitted two ROH

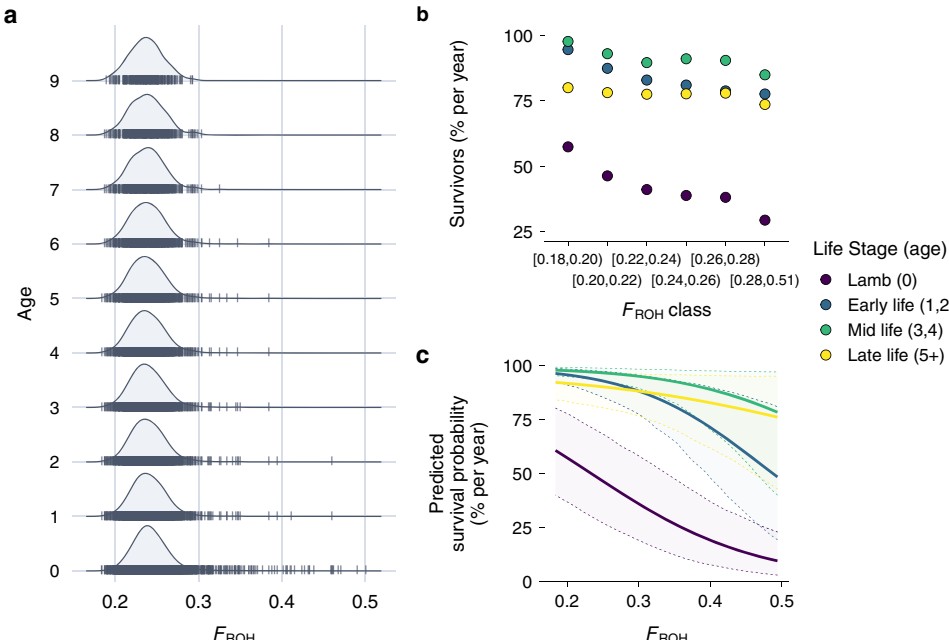

**Fig. 3 Inbreeding depression in annual survival. a** Distributions of inbreeding coefficients $F_{ROH}$ in Soay sheep age classes ranging from 0 to 9 years. **b** Proportion of surviving individuals per year in four different life stages and among different $F_{ROH}$ classes. As highly inbred individuals are relatively rare, the last class spans a wider range of inbreeding coefficients. Source data for this figure are also provided as a source data file. **c** Predicted survival probability and 95% credible intervals over the range of inbreeding coefficients $F_{ROH}$ for each life stage while holding sex and twin constant at 1 (male) and 0 (no twin). The predictions for the later life stages classes exceed the range of the data but are shown across the full range for comparability.

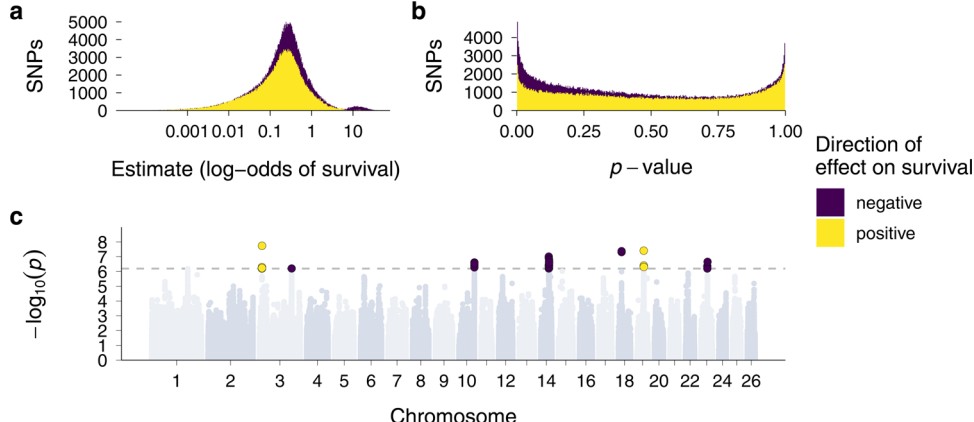

**Fig. 4 GWAS of SNP-wise ROH status effects on annual survival.** Regional inbreeding depression was conceptualised and tested using two binary ROH status predictors. One of the predictors quantified the ROH status of allele A (in ROH = 1, not in ROH = 0), while the other quantified the ROH status of allele B. **a** Distribution of effect sizes for SNP-wise ROH status effects. **b** Distribution of $p$-values for SNP-wise ROH status effects. The yellow histograms showing positive effects are superimposed on top of the purple histograms showing negative effects to highlight a substantially larger proportion of negative ROH status effects than expected by chance. **c** Manhattan plot of the ROH status $p$-values across the genome. The dotted line marks the genome-wide significance threshold for a Bonferroni correction which was based on the effective number of tests when accounting for linkage disequilibrium.

status predictors. The first predictor was assigned a 1 if allele A was homozygous and part of an ROH and a 0 otherwise. The second predictor was assigned a 1 if allele B was homozygous and part of an ROH and a 0 otherwise. Model estimates and $p$-values for these two predictors, therefore, reflect whether ROH are associated with survival consequences at each SNP location and for each allele. In the GWAS model, we also controlled for the additive SNP effect and mean individual inbreeding $F_{ROH}$ (based on all autosomes except for the focal chromosome), alongside a range of other individual traits and environmental effects (see 'Methods' section for details).

A GWAS on allele-wise ROH status can detect deleterious recessive alleles at specific regions when ROH effects reach

genome-wide significance. Moreover, the distribution of ROH status effects across the genome can also be informative of the overall number of deleterious recessive alleles contributing to inbreeding depression through ROH. Under the null hypothesis that ROH status does not have an effect on survival at any SNP position, we would expect a 50/50 distribution of negative and positive ROH status estimates due to chance. In contrast, we found many more negative than positive effects of ROH status on survival across the genome than expected by chance (Fig. 4a, b; 465 K neg. vs. 354 K pos.; exact binomial test $p = 2.2 * 10^{-16}$). Moreover, the proportion of negative ROH effects increases for larger model estimates (Fig. 4a) and smaller $p$-values (Fig. 4b). We tested this statistically using two binomial generalised linear

models (GLMs), with effect direction as a binary response, and model estimate and p-value, respectively, as predictors. ROH effects were more likely to be negative when their model estimate was larger (log-OR [95% CI] = 0.35 [0.344, 0.358]) and when their p-value was smaller (log-OR [95% CI] = −3.82 [−3.84, −3.80]). Consequently, it is likely that a large number of recessive deleterious alleles contribute to inbreeding depression, which manifests in negative ROH effects spread across many loci.

The GWAS revealed genome-wide significant ROH effects in seven regions on chromosomes 3 (two regions), 10, 14, 18, 19 and 23 (Fig. 4c and Supplementary Table 5). In five of these regions, ROH status for one of the alleles was associated with negative effects on survival, likely caused by relatively strongly deleterious recessive alleles. ROH in two further regions on chromosomes 3 and 19 were associated with increased survival probabilities, possibly due to haplotypes with positive effects on survival. To explore the genomic regions with large ROH effects further, we quantified the ROH density and SNP heterozygosity in 2 Mb windows around the top GWAS hits (Supplementary Fig. 8). Strongly deleterious recessive alleles might be expected to occur in regions of elevated heterozygosity where they are rarely expressed in their homozygous state. Heterozygosity was higher than average around the top SNPs on chromosomes 10, 14, 18 and 23, and ROH frequency was lower around the top SNPs on chromosomes 10, 18, 19 and 23, but overall, but we did not observe a convincing pattern of genetic diversity across the five regions harbouring strongly deleterious mutations (Supplementary Fig. 8).

## Discussion

The Soay sheep on St. Kilda have existed at a small population size in relative isolation for thousands of years[54]. As a consequence, levels of IBD are high in the population and ROH make up nearly a quarter of the average autosomal Soay sheep genome. Although this is still an underestimate as we only analysed ROH longer than 1.2 Mb, it is three times as high as the average $F_{ROH}$ estimated across 78 mammal species based on genome-sequence data[61] and only slightly lower than in some extremely inbred and very small populations such as mountain gorillas[62], Scandinavian grey wolves[24] or Isle Royale wolves[17].

The distribution of ROH length classes can provide insights into population history and levels of inbreeding[24,31,36]. In Soay sheep, the largest fraction of IBD was comprised of ROH with lengths between 1.2 and 4.9 Mb. These ROH originate from haplotypes around 8–32 generations ago and their relative abundance reflects a smaller $N_e$ of the population in the recent past. While there is considerably higher uncertainty in estimating the time to the MRCA for ROH when these are measured based on physical rather than genetic map lengths[12], this corroborates with historical knowledge: the Soay sheep population was indeed smaller in the early twentieth century when 107 sheep were translocated from the island of Soay to their current location on Hirta, after the last humans left St. Kilda[54]. Current levels of inbreeding were most visible in the variation in long ROH (>19.5 Mb), which made up <1% of the genome of the least inbred individuals on average, but 18% of the genome of highly inbred individuals with pedigree inbreeding coefficients $F_{ped} > 0.2$ (Supplementary Fig. 5).

The abundance of ROH in the population also varied substantially across the genome. In the most extreme ROH deserts and islands, only 4% of individuals and up to 87% of individuals had ROH, respectively, compared to 24% on average. These detected ROH islands and deserts could be regions with genuinely low or high levels of IBD due to genetic drift or natural selection, but they might also be a consequence of low or high

recombination rates[31]. However, recombination itself cannot change the true abundance of IBD[30]. Instead, ROH with a given coalescent time will be shorter in regions with high recombination and are less likely to be detected by ROH calling algorithms[30]. We modelled this and found that only 4% of the variation in ROH density across the genome was explained by variation in recombination rate, but the impact of recombination was greater on the detected densities of longer ROH. Consequently, the association between ROH density and recombination could change with both the minimum ROH threshold and the average inbreeding levels in a population. In line with the low genome-wide effects of recombination on ROH density, many of the ROH islands and deserts also had very similar recombination rates (Fig. 2c). Ruling out recombination as a major driver for ROH islands and deserts opens up the possibility for future studies to compare the extreme ROH density in islands and deserts to expectations under simulated neutral scenarios to test for positive and purifying selection, respectively[24,31].

ROH deserts might for example harbour loci contributing to inbreeding depression, as strongly deleterious alleles are likely to cause ROH to be rare in their genomic vicinity due to purifying selection removing homozygous haplotypes[7,24,31]. However, because of the near absence of ROH, genome-wide association analyses are unlikely to pick up deleterious effects due to a lack of statistical power, and indeed none of our top GWAS hits was located in a ROH desert. An alternative option is to test for deficits of homozygous genotypes under expected frequencies, as deployed in farm animals and plants to identify embryonic lethals[33–35], though these methods require either an experimental setup or very large sample sizes with tens to hundreds of thousands of individuals. In contrast, ROH islands with very high ROH abundances probably contain very few recessive deleterious alleles, as these are regularly exposed to selection when homozygous and hence likely to be purged from the population. Instead, it is possible that ROH islands have emerged around loci under positive selection[29,31] through hard selective sweeps[30].

We have only recently begun to understand the precise consequences of inbreeding for individual fitness in natural populations. In Soay sheep, we found that the odds of survival decreased by 60% with a mere 10% increase in $F_{ROH}$, adding to a small yet growing body of genomic studies reporting stronger effects of inbreeding depression in wild populations than assumed in pre-genomics times[13,14,50,51,53]. Other recent examples include life-time breeding success in red deer, which is reduced by up to 95% in male offspring from half-sib matings[13] and lifetime reproductive success in helmeted honeyeaters, which is up to 90% lower with a 9% increase in homozygosity[51]. The traditional way to compare inbreeding depression among studies is to estimate the inbreeding load of a population using lethal equivalents[59], although differences in methodology and inbreeding estimates can make such direct comparisons difficult[60]. We estimated the diploid number of lethal equivalents 2B for Soay sheep annual survival at 4.57 (95% CI 2.61–6.55). While this is a low to moderate inbreeding load compared to the few available estimates from wild mammals obtained from appropriate statistical models[60], none of these estimates are based on genomic data and they vary in their exact fitness measure as well as the degree to which they control for environmental and life-history variation. As such, average estimates of lethal equivalents might change in magnitude with the increasing use of genomics in individual-based long-term studies.

Inbreeding depression is dynamic across life, and genomic measures are starting to unravel how inbreeding depression affects fitness at different life stages in wild populations[13,14,50]. Under the mutation accumulation hypothesis[63], the adverse effects of deleterious mutations expressed late in life should become stronger as

selection becomes less efficient. Assuming mutation accumulation, inbreeding depression is expected to increase with age too[64], but empirical evidence is sparse[15,65,66]. In contrast, we showed that inbreeding depression in Soay sheep becomes weaker at later life stages. In addition, the sample for each successive age class consists of increasingly outbred individuals (Fig. 3a) due to a higher death rate among inbred individuals earlier in life. This suggests that the effects of intragenerational purging[67] outweigh mutation accumulation in shaping the dynamics of inbreeding depression across the lifetime.

The effect size distribution of loci underpinning inbreeding depression has to our knowledge not been studied in wild populations using fitness data, although deleterious mutations have been predicted from sequencing data, for example in ibex and Isle Royale wolves[17,18]. Theoretical predictions about the relative importance of weakly and strongly deleterious (partially-) recessive alleles will depend on many factors, such as the distribution of dominance and selection coefficients for mutations relative to the effective population size, and the frequency of inbreeding[68,69]. However, we could expect that small populations purge largely deleterious recessive mutations more efficiently as these are more frequently exposed to selection in the homozygous state[7,16,19,41], while weakly deleterious mutations can more often drift to higher frequencies. We estimated the effect of ROH status on Soay sheep survival for each of the two alleles at every SNP position within a GWAS framework. The effect size distribution revealed predominantly negative effects of ROH status on survival, particularly towards larger model estimates, showing that many alleles with weakly deleterious effects (or at low frequencies) probably contribute to inbreeding depression in survival.

Associations between ROH and survival reached genome-wide significance in seven regions on six chromosomes. In two of these regions, allele-specific ROH are predicted to increase survival, a fascinating observation we intend to explore in more detail but which is beyond the scope of this manuscript. In five further regions, ROH caused significant depression in survival, presumably due to loci harbouring strongly deleterious recessive alleles. This is unexpected, as Soay sheep have a long-term small population size with an estimated $N_e$ of 197[57], and strongly deleterious mutations should be rapidly purged. On the one hand, it is possible that genetic drift counteracted the effects of purifying selection and has allowed deleterious mutations to increase in frequency and be detected in a GWAS. On the other hand, a relatively recent admixture event with the Dunface sheep breed around 150 years ago[55] could have introduced deleterious variants into the population and recent selection has not been efficient enough to purge them from the population yet. Identifying the loci harbouring these strongly deleterious alleles will be challenging as ROH overlapping a given SNP vary in length among individuals, which makes it difficult to pinpoint an exact effect location. Nevertheless, we have shown that it is possible to identify the haplotypes carrying deleterious alleles with large effects. The frequencies of such haplotypes could be monitored in natural populations, and individuals carrying them could be selected against in conservation breeding programs. To sum up, our study shows how genome-wide marker information for a large sample of individuals with known fitness can deepen our understanding of the genetic architecture and lifetime dynamics of inbreeding depression in the wild.

## Methods

**Study population, pedigree assembly and survival measurements**. The Soay sheep (*Ovis aries*) is a primitive sheep breed descended from Bronze Age domestic sheep and has lived unmanaged on island of Soay in St. Kilda archipelago, Scotland for thousands of years. When the last human inhabitants left St. Kilda in 1932, 107

Soays were transferred to the largest island, Hirta, and have roamed the island freely and unmanaged ever since. The population increased and fluctuates nowadays between 600 and 2200 individuals. A part of the population in the Village Bay area of Hirta (57 49′N, 8 34′W) has been the subject of a long-term individual-based study since 1985[54]. Most individuals born in the study area (95%) are ear-tagged and DNA samples are obtained from ear punches or blood sampling. Routine mortality checks are conducted throughout the year with peak mortality occurring at the end of winter and beginning of spring. Overall, around 80% of deceased animals are found[50]. For the analyses in this paper, survival was defined as dying (0) or surviving (1) from the 1st May of the previous year to the 30th April of that year, with measures available for 5952 individuals from 1979 to 2018. Annual survival data were complete for all individuals in the analysis, as the birth year was known and the death year of an individual was known when it has been found dead during one of the regular mortality checks on the island. We focused on annual measures as this allowed us to incorporate the effects of age and environmental variation.

To assemble the pedigree, we inferred parentage for each individual using 438 unlinked SNP markers from the Ovine SNP50 BeadChip, on which most individuals since 1990 have been genotyped[70]. Based on these 438 markers, we inferred pedigree relationships using the R package Sequoia[71]. In the few cases where no SNP genotypes were available, we used either field observations (for mothers) or microsatellites[72]. All animal work was carried out in compliance with all relevant regulations for animal testing and research according to UK Home Office procedures and was licensed under the UK Animals (Scientific Procedures) Act of 1986 (Project License no. PPL70/8818).

**Genotyping**. We genotyped a total of 7,700 Soay on the Illumina Ovine SNP50 BeadChip containing 51,135 SNP markers. To control for marker quality, we first filtered for SNPs with minor allele frequency (MAF) > 0.001, SNP locus genotyping success >0.99 and individual sheep genotyping success >0.95. We then used the check.marker function in GenABEL version 1.8-0[73] with the same thresholds, including identity by state with another individual <0.9. This resulted in a dataset containing 39,368 polymorphic SNPs in 7700 sheep. In addition, we genotyped 189 sheep on the Ovine Infinium HD SNP BeadChip containing 606,066 SNP loci. These sheep were specifically selected to maximise the genetic diversity represented in the full population as described in Johnston et al.[58]. As a quality control, monomorphic SNPs were discarded, and SNPs with SNP locus genotyping success >0.99 and individual sheep with genotyping success >0.95 were retained. This resulted in 430,702 polymorphic SNPs for 188 individuals. All genotype positions were based on the Oar_v3.1 sheep genome assembly (GenBank assembly ID GCA_000298735.1[74]).

**Genotype imputation**. In order to impute genotypes to high density, we merged the datasets from the 50 K SNP chip and from the HD SNP chip using PLINK v1.90b6.12 with --bmerge[75]. This resulted in a dataset with 436,117 SNPs including 33,068 SNPs genotyped on both SNP chips. For genotype imputation, we discarded SNPs on the X chromosome and focused on the 419,281 SNPs located on autosomes. The merged dataset contained nearly complete genotype information for 188 individuals who have been genotyped on the HD chip, and genotypes at 38,130 SNPs for 7700 individuals who have been genotyped on the 50 K chip. To impute the missing SNPs, we used AlphaImpute v1.98[56], which combines information on shared haplotypes and pedigree relationships for phasing and genotype imputation. AlphaImpute works on a per-chromosome basis, and phasing and imputation are controlled using a parameter file (for the exact parameter file, see analysis code). Briefly, we phased individuals using core lengths ranging from 1 to 5% of the SNPs on a given chromosome over 10 iterations, resulting in a haplotype library. Based on the haplotype library, missing alleles were imputed using the heuristic method over five iterations which allowed us to use genotype information imputed in previous iterations. We only retained imputed genotypes for which all phased haplotypes matched and did not allow for errors. We also discarded SNPs with call rates below 95% after imputation. Overall, this resulted in a dataset with 7691 individuals, 417,373 SNPs and a mean genotyping rate per individual of 99.5% (range 94.8–100%).

To evaluate the accuracy of the imputation we used 10-fold leave-one-out cross-validation. In each iteration, we masked the genotypes unique to the high-density chip for one random individual that had been genotyped at high-density (HD) and imputed the masked genotypes. This allowed a direct comparison between the true and imputed genotypes. The imputation accuracy of the HD individuals should reflect of the average imputation accuracy across the whole population because HD individuals were selected to be representative of the genetic variation observed across the pedigree (see Johnston et al.[58] for details).

**ROH calling and individual inbreeding coefficients**. The final dataset contained genotypes at 417,373 SNPs autosomal SNPs for 5925 individuals for which annual survival data was available. We called runs of homozygosity (ROH) with a minimum length of 1200Kb and spanning at least 50 SNPs with the --homozyg function in Plink[75] and the following parameters: --homozyg-window-snp 50 --homozyg-snp 50 --homozyg-kb 1200 --homozyg-gap 300 --homozyg-density 200 --homozyg-window-missing 2 --homozyg-het 2 --homozyg-window-het 2. We chose

1200Kb as the minimum ROH length because between-individual variability in ROH abundance becomes very low for shorter ROH. Moreover, ROH of length 1200Kb extend well above the LD half decay in the population, thus capturing variation in IBD due to more recent inbreeding rather than linkage disequilibrium (Supplementary Fig. 9). The minimum ROH length of 1200Kb also reflects the expected length when the underlying IBD haplotypes had a most recent common ancestor haplotype 32 generations ago, calculated as $(100/(2*g))$ cM/1.28 cM/Mb where $g$ is 32 generations and 1.28 is the sex-averaged genome-wide recombination rate in Soay sheep[42,58]. To plot the ROH length distribution, we used the same formula to cluster ROH according to their physical length into length classes with expected MRCA ranging from 2 to 32 generations ago (Fig. 1b). Notably, ROH with the same physical length can have different coalescent times in different parts of the genome, causing a higher variance around the expected mean length than ROH measured in terms of genetic map length[12,42]. We then calculated individual inbreeding coefficients $F_{ROH}$ by summing up the total length of ROH for each individual and dividing this by the total autosomal genome length[43] (2452 Mb).

**ROH landscape and recombination rate variation.** To quantify variation in population-wide ROH density and its relationship with recombination rate and SNP heterozygosity across the genome, we used a sliding window approach. For all analyses, we calculated these estimates in 500 Kb non-overlapping sliding windows comparable to similar studies[24,30]; each window contained 85 SNPs on average. Specifically, we first calculated the number of ROH overlapping each SNP position in the population using PLINK --homozyg. We then calculated the mean number of ROH overlapping SNPs in 500 Kb non-overlapping sliding windows in the population (Fig. 1c). To estimate the top 0.5% ROH deserts and islands[31], windows with less than 35 SNPs (the percentile of windows with the lowest SNP density) were discarded. To estimate the impact of recombination rate on ROH frequency across the genome, we then quantified the recombination rate in 500 Kb windows using genetic distances from the Soay sheep linkage map[76]. Window heterozygosity was calculated as the mean SNP heterozygosity of all SNPs in a given window. Next, we constructed a linear mixed model in lme4[77] with population-wide ROH density (defined as the proportion of individuals with ROH) per window as response, window recombination rate and heterozygosity as fixed effects and chromosome ID as random intercept (Supplementary Table 3). The fixed effects in the model were standardised using $z$-transformation. We estimated the relative contribution of recombination and heterozygosity to variation in ROH density by decomposing the marginal $R^2$ of the model[78] into the variation explained uniquely by each of the two predictors using semi-partial $R^2$ as implemented in the partR2 package[79], with 95% confidence intervals obtained by parametric bootstrapping.

**Modelling inbreeding depression in survival.** We modelled the effects of inbreeding depression in annual survival using a Bayesian animal model in INLA[80]. Annual survival data consists of a series of 1 s followed by a 0 in the year of a sheep's death, or only a 0 if it died as a lamb, and we consequently modelled the data with a binomial error distribution and logit link. We used the following model structure:

$$\Pr(\text{surv}_i = 1) = \text{logit}^{-1}(\beta_0 + F_{ROH_i}\beta_1 + \text{earlyLife}_i\beta_2 + \text{midLife}_i\beta_3 + \text{lateLife}_i\beta_4 \\ + \text{sex}_i\beta_5 + \text{twin}_i\beta_6 + F_{ROH_i}\text{earlyLife}_i\beta_7 + F_{ROH_i}\text{midLife}_i\beta_8 \\ + F_{ROH_i}\text{lateLife}_i\beta_9 + \alpha_j^{\text{capture year}} + \alpha_k^{\text{birth year}} + \alpha_l^{\text{id}} + u_l^{\text{ped}})$$

(1)

$$\begin{aligned}
\alpha_j^{\text{capture year}} &\sim N(0, \sigma_{\text{year}}^2), & \text{for } j = 1, \dots, 40 \\
\alpha_k^{\text{birth year}} &\sim N(0, \sigma_{\text{birth year}}^2), & \text{for } k = 1, \dots, 40 \\
\alpha_l^{\text{id}} &\sim N(0, \sigma_{\text{id}}^2), & \text{for } l = 1, \dots, 5925 \\
u_l^{\text{ped}} &\sim N(0, A\sigma_A^2), & \text{for } l = 1, \dots, 5925
\end{aligned}$$

Here, $\Pr(\text{surv}_i = 1)$ is the probability of survival for observation $i$, which depends on the intercept $\beta_0$, a series of fixed effects $\beta_1$ to $\beta_9$, the random effects $\alpha$, which are assumed to be normally distributed with mean 0 and variance $\sigma^2$ and the breeding values $u_l^{\text{ped}}$, which have a dependency structure corresponding to the pedigree, with a mean of 0 and a variance of $A\sigma_A^2$, where $A$ is the relationship matrix and $\sigma_A^2$ is the additive genetic variance. We used a pedigree-derived additive relatedness matrix for computational efficiency as has previously been described for INLA models[81]. Variance component estimates for additive genetic effects have also previously been shown to be very similar when using a pedigree and an SNP-derived relatedness matrix in Soay sheep[70]. Also, additive genetic variance in Soay sheep fitness components is very low (Supplementary Table 4 and refs. [72,82]). As fixed effects, we fitted the individual inbreeding coefficient $F_{ROH}$ (continuous), the life stage of the individual (categorical predictor with four levels: lamb (age = 0, reference level), early life (age = 1, 2), mid-life (age = 3, 4), late-life (age = 5+)), sex (0 = female, 1 = male)), and a variable indicating whether an individual was born as a twin (0 = singleton, 1 = twin). We also fitted an interaction term of $F_{ROH}$ with life stage to estimate how inbreeding depression changes across the lifetime. As life stage was fitted as a factor, the model estimates three main effects for the differences in the log-odds of survival between lambs (reference category) and early life, mid-life and late-life respectively. Similarly, the interaction term between $F_{ROH}$ and life stage

estimates the difference in the effect of $F_{ROH}$ on survival (i.e. inbreeding depression) between lambs and individuals in early life, mid-life and late-life, respectively. As random intercepts, we fitted the birth year of an individual, the observation year to account for survival variation among years and the sheep identity to account for repeated measures. For all random effects, which are estimated in terms of precision rather than the variance in INLA, we used log-gamma priors with both shape and inverse-scale parameter values of 0.5. Supplementary Table 4 gives an overview over the coding and standardisation of the predictors.

Before modelling, we mean-centred[83,84] and transformed $F_{ROH}$ from its original range 0–1 to 0–10, which allowed us to directly interpret model estimates as resulting from a 10% increase in genome-wide IBD rather than the difference between a completely outbred and a completely inbred individual. Finally, we report model estimates as odds ratios in the main paper, and on the link scale (as log-odds ratios) in the Supplementary Material.

**Estimating lethal equivalents.** The traditional way of comparing inbreeding depression among studies is to quantify the inbreeding load in terms of lethal equivalents, i.e. a group of genes that would cause on average one death if dispersed in different individuals and made homozygous[59]. However, differences in statistical methodology and inbreeding measures make it difficult to compare the strength of inbreeding depression in terms of lethal equivalents among studies[60]. Here, we used the approach suggested by Nietlisbach et al.[60] and refitted the survival animal model with a Poisson distribution and a logarithmic link function. We were interested in lethal equivalents for the overall strength of inbreeding depression rather than its lifetime dynamics, so we fitted a slightly simplified animal model with $F_{ROH}$, age, age$^2$, twin and sex as fixed effects and birth year, capture year, individual id and pedigree relatedness as random effects. The slope of $F_{ROH}$ estimates the decrease in survival due to a 10% increase in $F_{ROH}$, so we calculated the number of diploid lethal equivalents 2B as $-(\beta_{F_{ROH}})/0.10 * 2$ where $\beta_{F_{ROH}}$ is the Poisson model slope for $F_{ROH}$.

**Mapping inbreeding depression.** To map the effects of inbreeding depression in survival across the genome, we used a modification of a genome-wide association study[48,49,85]. For each of the ~417 K SNPs, we fitted a binomial mixed model with logit link in lme4[77] with the following model structure:

$$\Pr(\text{surv}_i = 1) = \text{logit}^{-1}(\beta_0 + \text{SNP}_{ROH_{\text{alleleA}_i}}\beta_1 + \text{SNP}_{ROH_{\text{alleleB}_i}}\beta_2 + \text{SNP}_{\text{ADD}_i}\beta_3 \\ + F_{ROH_{\text{mod}_i}}\beta_4 + \text{age}_i\beta_5 + \text{age}_i^2\beta_6 + \text{sex}_i\beta_7 + \text{twin}_i\beta_8 \\ + \beta_{9-16}\text{PC}_{1-7} + \alpha_j^{\text{capture year}} + \alpha_k^{\text{birth year}} + \alpha_l^{\text{id}})$$

(2)

$$\begin{aligned}
\alpha_j^{\text{capture year}} &\sim N(0, \sigma_{\text{capture year}}^2), & \text{for } j = 1, \dots, 40 \\
\alpha_k^{\text{birth year}} &\sim N(0, \sigma_{\text{birth year}}^2), & \text{for } k = 1, \dots, 40 \\
\alpha_l^{\text{id}} &\sim N(0, \sigma_{\text{id}}^2), & \text{for } l = 1, \dots, 5925
\end{aligned}$$

Where the effects of interest are the two ROH status effects, $\text{SNP}_{ROH_{\text{alleleA}}}$ and $\text{SNP}_{ROH_{\text{alleleB}}}$. These are binary predictors indicating whether allele A at a given SNP is in an ROH ($\text{SNP}_{ROH_{\text{alleleA}}} = 1$) or not ($\text{SNP}_{ROH_{\text{alleleA}}} = 0$), and whether allele B at a given SNP is an ROH ($\text{SNP}_{ROH_{\text{alleleB}}} = 1$) or not ($\text{SNP}_{ROH_{\text{alleleB}}} = 0$). These predictors test whether an ROH has a negative effect on survival, and whether the haplotypes containing allele A or the haplotypes containing allele B are associated with the negative effect and therefore carry the putative recessive deleterious mutation. $\text{SNP}_{\text{ADD}}$ is the additive effect for the focal SNP (0, 1, 2 for homozygous, heterozygous, homozygous for the alternative allele, respectively), and controls for the possibility that a potential negative effect of ROH status is simply an additive effect. $F_{ROH_{\text{mod}}}$ is the mean inbreeding coefficient of the individual based on all chromosomes except for the chromosome where the focal SNP is located. We fitted $F_{ROH}$ as we were interested in the effect of ROH status at a certain locus on top of the average individual inbreeding coefficient. Sex and twin are again binary variables representing the sex of the individual and whether it is a twin. Age and age$^2$ control for the linear and quadratic effects of age, and are fitted as continuous covariates. Because it was computationally impractical to fit 417 K binomial animal models with our sample size and because the additive genetic variance in survival was very small (posterior mean [95% CI] = 0.29 [0.22, 0.37], see Supplementary Table 4), we did not fit an additive genetic effect. Instead, we used the top 7 principal components of the variance-standardised additive relationship matrix ($\text{PC}_{1-7}$) as fixed effects[75]. Again, we added birth year, capture year and individual id as random effects. For each fitted model, we extracted the estimated slope of the two $\text{SNP}_{ROH}$ predictors and their $p$-values, which were calculated based on a Wald-Z test. To determine a threshold for genome-wide significance we used the 'simpleM' procedure[86]. The method uses composite linkage disequilibrium to create a SNP correlation matrix and calculates the effective number of independent tests, which was much lower than the number of SNPs ($n_{\text{eff}} = 39184$) because LD stretches over large distances in Soay sheep (Supplementary Fig. 9). We then doubled this number to 78368, as we conducted two tests per model, and used this value for a Bonferroni correction[87] of $p$-values, resulting in a genome-wide significance threshold of $p < 6.38 * 10^{-7}$.

**Reporting summary**. Further information on research design is available in the Nature Research Reporting Summary linked to this article.

## Data availability

All data are available size on Zenodo (https://doi.org/10.5281/zenodo.4609701)[88]. Source data are provided with this paper.

## Code availability

Code was written in R 3.6.1[89] and relied heavily on tidyverse 1.3.0[90], data.table 1.14.0[91] and ggplot2 3.3.3[92]. The full analysis code is available on Zenodo (https://doi.org/10.5281/zenodo.4587676)[93] and GitHub (https://github.com/mastoffel/sheep_ID).

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

## Acknowledgements

We thank the National Trust for Scotland for permission to work on St. Kilda and QinetiQ, Eurest and Kilda Cruises for logistics and support. We thank Ian Stevenson and many volunteers who have helped with data collection and management and all those who have contributed to keeping the project going. SNP genotyping was conducted at the Wellcome Trust Clinical Research Facility Genetic Core. This work has made extensive use of the Edinburgh Compute and Data Facility (http://www.ecdf.ed.ac.uk/). We thank John Hickey, Steve Thorn, Andrew Whalen and Martin Johnsson for help with the imputation. We are grateful for discussions with David Clark, Holger Schielzeth, Matthias Galipaud, Jon Slate, Jisca Huisman, Jarrod Hadfield, Peter Visscher, Anna Hewett, Deborah Charlesworth, Brian Charlesworth and the Wild Evolution Group. We also thank Emily Humble for comments on an earlier draft of the manuscript. The project was funded through an outgoing Postdoc fellowship from the German Science Foundation (DFG) awarded to M.A.S. and a Leverhulme Grant (RPG-2019-072) awarded to J.M.P. and S.E.J. Field data collection has been supported by NERC over many years, and most of the SNP genotyping was supported by an ERC Advanced Grant to JMP.

## Author contributions

J.M.P. and M.A.S. designed the study. J.G.P. is the main Soay sheep project fieldworker and collected samples and life-history data. J.M.P. has run the long-term Soay sheep study and organised the SNP genotyping. S.E.J. built the core genomic database, including genotyping, quality control and linkage mapping. M.A.S. conducted data analyses and drafted the manuscript. M.A.S., J.M.P. and S.E.J. jointly contributed to concepts, ideas and revisions of the manuscript.

## Competing interests

The authors declare no competing interests.
