## [Peer Review File · Nature Communications]

REVIEWER COMMENTS

Reviewer #1 (Remarks to the Author):

Genetic architecture and lifetime dynamics of inbreeding depression in a wild mammal

This manuscript addresses a fundamental concept in evolutionary and conservation biology: inbreeding depression in fitness traits. Quantifying inbreeding depression in fitness traits in natural populations is not easy because it requires long-term data collection in the field and extensive genomic work. This is challenging and hence few estimates are available. This manuscript is therefore a very welcome addition to the growing body of evidence on inbreeding depression in fitness traits.

This study is based on an amazing data set with data from 5942 individuals and 417k SNPs. Severe inbreeding depression is evident in this data set: a 10% increase in inbreeding reduces annual survival on average by 68%. However, the magnitude of the inbreeding effects varies with age of the animals. It is lowest in lambs and most pronounced in young adults. A genome-wide association study furthermore revealed that many of the loci involved have a minor effect on fitness. However, a handful of loci seems to be strongly deleterious.

Overall this is a very nice contribution to the field and in addition to a number of detailed comments below I only have three main remarks that I hope will improve the manuscript:

1) I think it would help readers not intimately familiar with the subject if the introduction gave more of a background why some of these questions are important. For example: Why is it important to understand how inbreeding causes IBD along the genome (lines 50/51 and 64). And why is the distribution of effect sizes at loci causing inbreeding depression important (line 52/53)? Clearly outlining the background for the questions posed would help readers see the big picture more sharply.

2) I was surprised that the distribution of ROH lengths receives hardly any attention. ROHs of different lengths represent inbreeding from more or less distant ancestors and thus ROHs of different lengths are thought to be differently enriched for homozygous deleterious coding variation based on the time to the most recent common ancestor and the degree of admixture (e.g. Szpiech et al. *Am. J. Hum. Genet.* 2019). Since the Soay sheep experienced long times of inbreeding and recent admixture, the distribution of ROH lengths should be presented and investigated.

3) I found it difficult to understand the way the authors chose to model age effects on inbreeding depression. Supplementary Table 4 indicates that the effect of being a lamb (age=0) was modelled with a categorical effect. In addition, the effect of age was modelled as a linear function of age, starting with age=0. Thus, the effect on survival of being a lamb is represented in two covariates (lamb and age), making it difficult to interpret the interaction between inbreeding and age. I think this issue may be at the heart of the discrepancy between the results given in the main text and Figure 2C. According to the main text, lambs experienced weaker inbreeding depression. But Figure 2C shows that survival of lambs

decreased much more with increasing inbreeding than survival of 1, 4, and 7-year-olds at least up to a F_{ROH} of 0.4. Given that an F_{ROH} of more than 0.4 is very rare in animals aged one and two and entirely absent in animals aged 3 and older, how can we interpret the main text in the light of Figure 2C?

Also, Supplementary Figure 5 suggests that a linear function as used in the inbreeding depression model provides a poor fit for the observed interaction between age and F_{ROH} . Interestingly, in the other models (lethal equivalents and GWAS) age was included in a non-linear fashion. Why was this not done in the inbreeding depression model? All in all, I think it would make the biological interpretations of the model more straight forward if age was modelled non-linearly as in the other models or with a categorical variable.

Detailed comments:

lines 50-51: The first question here (“What are the genomic patterns of homozygosity caused by inbreeding?”) seems very specific compared to the much more general questions that follow. Also, at this stage of the manuscript the reader has no clue why it is important to know the genomic patterns of homozygosity. Please clarify.

lines 66-77: I found this introduction to runs of homozygosity a little superficial. There is no mention of the dependence of the length of ROH on the time to the next common ancestor, no mention of the effects of the demographic history of a population (except through a somewhat oblique reference to ‘background relatedness’), and no mention of the effects of selection. Please provide some more background here.

line 87: An additional study that probably appeared after the authors submitted this manuscript is Niskanen et al. 2020, PNAS (Consistent scaling of inbreeding depression in space and time in a house sparrow metapopulation).

lines 111-112: Please specify in the Methods what ‘complete annual survival and genomic data’ implies. Does it imply that you only used animals that were found dead (as indicated in the line above) and from cohorts where everyone had died by the end of the study? And what settings were used to define complete genomic data?

line 115: Remove one ‘on’.

line 134: How can the mean numbers of ROH in these two subsets (183 and 174) be lower than in the complete dataset (194)?

line 138: Please explain (possibly in the Methods) why you used 500Kb windows.

Figure 1B: Please provide and label the y axis of the density plot insert.

lines 209-210: It did not become clear to me how you arrived at this conclusion. Please explain.

lines 210-212: How long, on average, were the ROHs that these three SNPs were in and how do the lengths compare to the distribution of ROH lengths in the population?

line 248: Large and small are relative terms. Hence, it would be good to say that here 'very large sample sizes' refers to more than 100'000 individuals.

lines 251-264: Taken at face value, this paragraph contradicts itself. The paragraph starts by pointing out how this and other genomic studies have revealed more inbreeding depression than previously thought. But it ends with the statement that this study revealed low to moderate inbreeding load. Many readers will likely find this contradictory. It would be important to clarify that genomic studies have found more inbreeding depression than studies of the same species that had used incomplete pedigrees. But that in comparison to other species the inbreeding load among Soay sheep is low to moderate. So, one is a technical reason the other a biological reason.

lines 261-264: This is true only if genetic load is considered the same as inbreeding load. Drift load would show a very different pattern. Since different authors define genetic load differently, it would be important to clearly define it here.

line 270: Keller et al. 2008, Proc. Roy Soc. provide additional but mixed evidence for increased inbreeding depression later in life.

lines 292-293: I did not understand what was meant by 'particularly towards larger effect sizes'. Please clarify.

line 338: Ref. 63 used a MAF cutoff of 0.01. Did you really use a cutoff of 0.001 here?

lines 341 and 345: The main text refers to 189 individuals genotyped with the high density SNP (line 116). Please clarify.

lines 376-378: How do these parameters compare to the recommendation of Meyermans et al. 2020, BMC Genomics?

lines 386-387: Please provide a reference.

line 393: I believe the 0.5% is based on the study by Pemberton et al. 2012? If this is correct, please cite this reference here.

line 421 and Supplementary Table 4: Please justify why you used a pedigree-derived additive

relatedness matrix in the animal models rather than a genomic relatedness matrix. The latter would seem more natural given the use of ROH as a measure of inbreeding (see e.g. Luan et al. 2014, *Genetics Selection Evolution*).

line 433: Please provide a reference for the choice of priors.

line 477: I would prefer a quantitative statement here instead of 'negligible'. Supplementary Table 4 allows such a quantitative statement.

line 498: Is an 'at' or 'by' missing in this sentence?

Reference list: The citations need some attention. Journal names and abbreviations, the use of upper and lower case, and the use of page numbers are inconsistent. For some references information is missing (e.g. ref 23).

Supplementary Figure 5: What are the error bars? 95% CIs?

Supplementary Table 1: Please indicate in the caption what N(Snps) stands for.

Reviewer #2 (Remarks to the Author):

Summary:

In this study, Stoffel et al. quantified the impact of inbreeding on annual survival in 5,952 wild Soay sheep. Their analyses included 1) identification of ROH in each individual based on imputed SNP genotypes at 417 thousand loci, 2) modeling the effect of Froh on annual survival in a Bayesian framework, and 3) a modified genome-wide association study to reveal the genetic architecture of inbreeding depression. They report evidence of inbreeding depression caused by many recessive deleterious mutations contained in large runs of homozygosity. They find that annual survival is reduced by 68% in an individual with Froh 10% above the population-wide mean, and that this reduced survival primarily manifests in the first few years of life.

Evaluation:

This is an impressive study because it combines a comprehensive genomic analysis with detailed life history data in a large number of samples collected over a period of several decades. This is extremely difficult to do in a wild population, and such studies remain rare. With this sample size and density of genomic data, the results are quite convincing. The methods are robust and detailed sufficiently, and I was very pleased to see that all of the analysis code has been made accessible. The manuscript is clear, well-organized, and cites appropriate literature. The results are interesting and add to our understanding of the genomic basis of inbreeding depression. In my opinion, the manuscript is of sufficient quality and novelty to interest the diverse readership of *Nature Communications*. I have only minor suggestions for the authors, as described below.

Minor suggestions:

- A figure showing the distribution of minor allele frequencies would be a helpful addition to the supplement. A mean minor allele frequency of 23% is very high, though perhaps not unexpected here due to the fact that the population is small, inbred, isolated, founded from ~100 individuals in the 1930s, etc. These factors make the population rather unique, and probably influence the genetic architecture of inbreeding depression in this system (i.e. few large effect mutations, evidence of many weakly deleterious recessive loci across the genome), which may mean the results would not hold generally in other populations with different demographic histories.
- It would be interesting to know whether Froh has changed over the course of the study period. Is there evidence of any change over the years or over the generations within the pedigree? If so, it would be interesting to note, and if not, it would provide more confidence in the results.
- Showing the survival probability curves (as in Fig. 2C) for the other age classes (years 2, 3, 5, 6, 8+) in the supplement would aid transparency.
- 268: The relevance of mutation accumulation in this system is not clear to me. I am not sure that the gradual build up of new mutations over time in the Soay population due to long-term reduced inefficiency of selection is the right framework for understanding inbreeding depression within this population. The literature on mutation accumulation is certainly important, but the subsequent two paragraphs (282-310) about the distribution of dominance and deleterious mutation effect sizes seem far more relevant to understanding the nature of genetic load and inbreeding depression in this system.
- 344: Was the same MAF filter used on the high density chip, which was run on far fewer individuals? MAF of .001 in a sample of 188 individuals is less than 1 allele.
- 432-433: Should this be referring to the fixed effects, not the priors on the random effects?
- Supplemental figure 7: A color-coded legend would be helpful here. The line colors are not explicitly mentioned in the caption.

Reviewer #3 (Remarks to the Author):

This is a very interesting paper. It is certainly one of the most comprehensive, if not the most comprehensive, analysis of inbreeding depression in a wild population to date. The paper looks at many of the key components of inbreeding and inbreeding depression: effects of inbreeding on a key fitness component, lethal equivalents, the genomic distribution of ROH density, and it identifies genomic regions that likely contribute to inbreeding depression using association analysis and a purely genomic approach. The methods section is comprehensive, and the interpretation of results is appropriate.

I strongly support publication of this paper in any journal. However, I suggest addressing some important issues in the paper beforehand. My comments are below. The most important are marked with an asterisk. In particular, I suggest digging deeper into the mechanisms leading to the high variance in the distribution of ROH density across the genome, and considering a modification to the association

analysis to increase statistical power and the accuracy of estimated locus-specific effect sizes on fitness.

68-70: It would be more precise and succinct to say that short ROH originate from distant ancestors (i.e., common relatives of the parents many generations ago).

131: What is the motivation for using 1.2 Mb as a minimum length threshold?

132-134: I find the wording awkward and had to read this a few times to figure out the meaning. You could instead write something like “the seven most inbred sheep had longer ROH (mean: 5.27 Mb vs. 2.76 Mb), and slightly more abundant ROH (mean: 183 vs. 174) than the seven least inbred sheep”

139-146: I wonder if the regions with exceptionally few ROH occur in regions that also have high admixture from Dunface. Do you have a map of the abundance of Dunface haplotypes across the Soay genome?

*142-146 and elsewhere relating to ROH abundance and recombination rate: This is very interesting and important, and the field would benefit from digging more deeply into the reasons for this pattern. The relationship between ROH density and recombination rate is substantially more complicated than it appears by reading the relevant literature. First, it seems worth putting a figure showing something like that in Fig S4 in the main text – for example, a single plot showing the data from all chromosomes combined at once. Then I suggest adding some material on the potential mechanisms underlying this pattern.

The main finding here is more ROH in regions with lower recombination rate. In lines 237-239, the potential influence of recombination rate, selection, and gene density are rightly pointed out. I agree with the proposed potential selective explanations: ROH deserts are good candidates for harboring strongly deleterious recessive alleles or loci under balancing selection; ROH islands may well be regions where selective sweeps have occurred. This analysis and argument would be strengthened by seeing if there are any ROH deserts in regions of the genome without exceptionally high recombination rate. In such cases, it would be hard to point to something related to recombination rate (see next paragraph) as the most likely explanation. However, the ROH islands and deserts might all be located in regions represented in the far left and right ends of the x-axes in Figure S4, respectively, in which case it is more difficult to point to selection as the most likely explanation. This is particularly important when ROH density is clearly associated with recombination rate (this was not the case in the cited Scandinavian wolf paper on ROH). I suggest investigating this, albeit briefly, at least as presented in the main text.

An issue with interpretation of ROH results that is particularly prevalent in the human genetics literature, and relevant here I believe, is that recombination by itself (i.e., without an additional influence of selection) cannot affect the TRUE abundance of ROH. Regions with higher recombination rate produce shorter ROH than parts of the genome with low recombination rate. This means it is generally easier to detect ROH in regions with low recombination because such ROH are more ‘obvious’ to ROH detection algorithms. This can by itself create the appearance of a correlation between ROH

density and recombination rate. We looked at this in detail in our paper on genomic analysis of inbreeding in flycatchers (Kardos et al. 2017, Genetics). See in particular Figure 7 in that paper, which shows that the density of DETECTED ROH can vary with recombination rate across genomes where inbreeding (heterozygosity) is really held constant across the genome. The issue is that ROH with a given coalescent time will be tiny in regions with very high recombination rate, and huge in regions with very low recombination rate. This phenomenon likely explains at least part of the genome-wide variation in ROH density here... You can easily detect big ROH in regions with low recombination rate, whereas it is harder to detect them in regions with high recombination rate because they are so short.

My suggestion is to at least raise this in the Discussion. Even better would be to test whether this is likely to explain the results by seeing if ROH density is correlated with heterozygosity (proportion of het SNPs in the same windows as ROH density is analyzed); if this association is significant, it would suggest that ROH abundance is indeed tracking variation in true total IBD across the genome. One caveat is that ascertainment bias associated with the construction of the SNP chips might make this difficult (genome sequence data would be better), but I am not sure. Another caveat is that with 'only' < 500K SNPs, one cannot account for the contribution of very short ROH.

*195-216: This is a very nice analysis and I am really glad to see it done. I particularly appreciate that it controls for genome-wide identity disequilibrium (469-470). However, there might be a missed opportunity to identify more loci contributing to inbreeding depression, and to precisely estimate their fitness effects. Consider a diallelic locus where allele A has a strong deleterious, mostly recessive effect on fitness, and allele B causes no inbreeding depression. The analysis here assumes that ROH presence/absence affects fitness equally regardless of allele/haplotype identity. This means that the fitness effects of homozygosity for the two alleles are essentially averaged in the analysis, if I understand the model correctly. In our hypothetical example, the present analysis would have perhaps substantially underestimated the effect size. Treating ROH status for each allele separately (i.e., test for an effect of ROH presence/absence separately for each allele) could in principle greatly increase power, and provide a much better estimate of locus/allele-specific fitness effect. An alternative, if the data are phased, would be to conduct the association test on haplotype-specific ROH status.

233-234: This figure would inevitably be considerably higher if you could detect shorter ROH (e.g., with whole genome resequencing). It would be worth mentioning this in the Discussion.

382-385: What is the reason for the threshold of 32 generations to the common ancestor haplotype? Measuring inbreeding due only to recent ancestors is much better done by defining the length threshold in terms of cM rather than Mb, specifically because the recombination rate obviously varies a lot across the genome. The way the analysis was done means that ROH of the same physical length (say 1.2 Mb) have very different average coalescent times in different parts of the genome; the time horizon therefore is quite 'fuzzy' in the analyses. I don't think this is a big problem. You could either redo the analysis using a cM-based threshold, or acknowledge that the realized g (line 384) must have a much higher variance across detected ROH of a given length class than would be the case with a cM-defined threshold.

483-484: I suggest reporting the effective number of tests.

488: If the data will be embargoed then I suggest stating that here.

163 and Figure 2b: The point estimate is reported as a 68% reduction in the text, but in the report of model estimates (in the parentheses), $1 - 0.68 = 0.32$ and its CI is given. Then the figure legend combined with Figure 2b suggests that the reduction in survival odds associated with a 0.1 increase in Froh is ~ 0.32 . First, I think the meaning of the parameter being estimated should be stated much more clearly. Additionally, the parameter estimate should be stated consistently everywhere it is mentioned (not as 0.68, and then $1 - 0.68$ elsewhere). It is hard to follow as written, and this would likely lead to some misinterpretation among readers. It is not clear to me which is correct.

263: Should this not be 'inbreeding load' instead of 'genetic load'.

(signed)

Marty Kardos

We'd like to thank all three reviewers for very thorough and constructive comments. The revised MS takes all comments into account and includes several new analyses, figures, and substantial changes on the manuscript.

REVIEWER COMMENTS

Reviewer #1 (Remarks to the Author):

Genetic architecture and lifetime dynamics of inbreeding depression in a wild mammal

This manuscript addresses a fundamental concept in evolutionary and conservation biology: inbreeding depression in fitness traits. Quantifying inbreeding depression in fitness traits in natural populations is not easy because it requires long-term data collection in the field and extensive genomic work. This is challenging and hence few estimates are available. This manuscript is therefore a very welcome addition to the growing body of evidence on inbreeding depression in fitness traits.

This study is based on an amazing data set with data from 5942 individuals and 417k SNPs. Severe inbreeding depression is evident in this data set: a 10% increase in inbreeding reduces annual survival on average by 68%. However, the magnitude of the inbreeding effects varies with age of the animals. It is lowest in lambs and most pronounced in young adults. A genome-wide association study furthermore revealed that many of the loci involved have a minor effect on fitness. However, a handful of loci seems to be strongly deleterious.

Overall this is a very nice contribution to the field and in addition to a number of detailed comments below I only have three main remarks that I hope will improve the manuscript:

Thank you for this and also for the detailed review.

1) I think it would help readers not intimately familiar with the subject if the introduction gave more of a background why some of these questions are important. For example: Why is it important to understand how inbreeding causes IBD along the genome (lines 50/51 and 64). And why is the distribution of effect sizes at loci causing inbreeding depression important (line 52/53)? Clearly outlining the background for the questions posed would help readers see the big picture more sharply.

R1: We have now rewritten and restructured the introduction and stated the questions more broadly, which we hope sets up the paper better.

2) I was surprised that the distribution of ROH lengths receives hardly any attention. ROHs of different lengths represent inbreeding from more or less distant ancestors and thus ROHs of different lengths are thought to be differently enriched for homozygous deleterious coding variation based on the time to the most recent common ancestor and the degree of admixture (e.g. Szpiech et al. Am. J. Hum. Genet. 2019). Since the Soay sheep experienced

long times of inbreeding and recent admixture, the distribution of ROH lengths should be presented and investigated.

R2: The distribution of ROH lengths is indeed an interesting topic, which we (consciously) omitted in our first draft of the paper. The main reason for this is that we felt that a thorough analysis of the relationship between ROH length classes and deleterious alleles is more difficult than it seems from the literature. To be properly addressed, we think that these analyses would need forward genetic simulations, their own set of mixed models and likely also a different sample than the one we chose for this manuscript. Specifically, it is not clear what we would actually expect, i.e. do we expect longer ROH to be enriched for deleterious alleles or not? This is also mentioned in the original paper addressing this issue (Szpiech et al., 2013) where the two working hypotheses are actually going in two entirely different directions. Moreover, while Szpiech et al. 2013 and 2019 used *predicted* deleterious alleles as evidence that longer ROH are enriched for deleterious variants, Clark et al. test this hypothesis on a large human sample with actual phenotypes (Clark et al., 2019) and don't find a difference in depression when comparing inbreeding coefficients based on ROH > 5Mb and ROH < 5Mb. Another study in cattle found the opposite pattern: short ROH were more enriched for deleterious variants than long ROH (Zhang, Guldbbrandtsen, Bosse, Lund, & Sahana, 2015). As a consequence, we are planning a separate manuscript, in which we will use forward genetic simulations to establish baseline expectations for the Soay sheep population. Moreover, to test these hypotheses, we would also focus on a different sample of individuals (e.g. only lambs), as inbred individuals die early and the distribution of ROH lengths therefore shifts across the lifetime.

Given these considerations, we decided that an in-depth investigation of the relationship between ROH length classes and inbreeding depression falls outside the scope of this manuscript. However, we appreciate that the ROH length distribution is of general interest and we now included a new Figure 1B showing the distribution of ROH across different length classes in the population. We also extended the introduction with new parts on ROH lengths (lines 72-78), and added new sections to the results (lines 141-147) and discussion (lines 312-322), qualitatively describing the relationship between the ROH length distribution, inbreeding and demography in the Soay sheep population.

3) I found it difficult to understand the way the authors chose to model age effects on inbreeding depression. Supplementary Table 4 indicates that the effect of being a lamb (age=0) was modelled with a categorical effect. In addition, the effect of age was modelled as a linear function of age, starting with age=0. Thus, the effect on survival of being a lamb is represented in two covariates (lamb and age), making it difficult to interpret the interaction between inbreeding and age.

R3: Yes, this is correct, the lambs at the age of 0 are represented as part of a continuous age variable and in addition as a binary covariate (0 = not lamb, 1 = lamb).

Although this is rarely done (but see for example section 2.8, last paragraph in Chakrabarty, van Kronenberg, Toliopoulos, & Schielzeth, 2019), it is a precise way to model two specific hypotheses simultaneously: (1) Is inbreeding depression becoming (linearly) weaker (or stronger) through life? And (2): Is inbreeding depression weaker (or stronger) in lambs? By fitting $\text{age} + \text{age:Froh} + \text{lamb} + \text{lamb:Froh}$, we estimate a slope for the linear trend of inbreeding depression through life (age:Froh). However, we hypothesise that a specific life stage (lamb) differs from this general linear trend. By also fitting lamb:Froh , we allow the model to estimate the interaction of inbreeding depression with lambs separately. This estimate is therefore allowed to independently deviate from the general linear trend across life in the model, which makes it possible to address a specific hypothesis for a particular age class (lambs) while still estimating a trend across all age classes.

A related question from the reviewer, R5 below, is why we did not fit age^2 as in the other models. To somehow (though not exactly) address our hypothesis we would need to fit the model $\text{age} + \text{age:Froh} + \text{age}^2 + \text{age}^2:\text{Froh}$. We would still need to include age as a main effect, because age^2 is an interaction of age with itself, and as such needs a main effect in the model to be interpreted properly, like other interactions too (Schielzeth, 2010). However, the question would become less specific, i.e. we would ask whether inbreeding depression is weaker (or stronger) for both lambs and old individuals. But we have many more lambs in the data than we have old individuals, and this would likely drive the estimated patterns anyway. As such, we think that our way of modelling this is correct and addresses our specific hypotheses. However, in the GWAS and lethal equivalents models, we simply want to control for age effects and are not specifically interested in interpreting them. We therefore fit age and age^2 as these predictors are likely to capture the relevant variation more generally. We now extended our explanations of the models in the methods section in lines 545 ff. to explain this more clearly.

I think this issue may be at the heart of the discrepancy between the results given in the main text and Figure 2C. According to the main text, lambs experienced weaker inbreeding depression. But Figure 2C shows that survival of lambs decreased much more with increasing inbreeding than survival of 1, 4, and 7-year-olds at least up to a F_{ROH} of 0.4. Given that an F_{ROH} of more than 0.4 is very rare in animals aged one and two and entirely absent in animals aged 3 and older, how can we interpret the main text in the light of Figure 2C?

R4: The model interpretation is indeed slightly tricky with respect to this specific result. This is due to the non-linear relationship between the log-odds estimates of the logistic regression model and the survival probability, which is calculated through an inverse logit transformation of the log-odds (or logit) estimates. If we plotted the same Figure 2C with the (untransformed) linear log-odds estimates the difference in slopes between age 0 and 1 is much clearer (see the new Supplementary Figure 8 or Rebuttal Figure 1 below).

Rebuttal Figure 1: Predicted survival on the log-odds scale. Unlike survival predictions on the probability scale, the model prediction on the log-odds scale are untransformed. As untransformed predictions are linear, the steeper slope and therefore stronger inbreeding depression at age 1 compared to lambs at age 0 is more apparent.

So why is the difference in slopes less clear when using the transformed survival probabilities? This is because the transformation is non-linear and depends on the 'baseline' survival probability. The survival probability for lambs is much lower than for adults, because we specifically modelled this using the "lamb" main effect. This fits well with our empirical data, where more than half of all lambs die over their first winter, thus having a substantially lower survival probability than adults. If we compare the slope of the curves at similar survival probabilities (i.e. 50%), the slope of age=1 is indeed much steeper than age=0. However, interpreting the raw data in conjunction with Figure C at face value, it is true that the survival curve in individuals between Froh=0.2 and Froh=0.3 is less steep in age 1 individuals, though this trend reverses for inbred individuals with Froh>0.3, where lambs are predicted to have weaker inbreeding depression.

In summary, this shows that the interpretation of this particular logistic regression model (and logistic regression models in general) is not entirely straightforward. We are quite confident that the direct interpretation of the model is correct when based on the untransformed estimates, and here the Froh*Lamb interaction predicts weaker inbreeding depression in lambs (Supplementary Table 4 and Figure 2B). However, we appreciate both the uncertainty in the estimate itself (large credible intervals) as well as the uncertainty in the prediction when transformed to the probability scale. We are transparent about these results in showing raw data alongside the model estimates and in providing the data and code necessary to reproduce them. As we feel that this result, though not rock solid, is very interesting, and because it is only a minor part of the current manuscript, we keep reporting it but toned down its interpretation and gave an extended explanation on the matter in the results (lines 214 ff.).

Also, Supplementary Figure 5 suggests that a linear function as used in the inbreeding depression model provides a poor fit for the observed interaction between age and F_{ROH} . Interestingly, in the other models (lethal equivalents and GWAS) age was included in a non-linear fashion. Why was this not done in the inbreeding depression model? All in all, I think it would make the biological interpretations of the model more straight forward if age was modelled non-linearly as in the other models or with a categorical variable.

R5: We do not agree that age as a linear function is a poor fit. While the inbreeding depression estimates shown in Supplementary Figure 5 are estimated through one model per age class, and thus represent a sub-optimal modelling strategy, we think that the figure reflects very much what our results show: A linear decrease of inbreeding depression with increasing age, with the exception of very young individuals, for which inbreeding depression is weaker. We refer to response 4 for the question of why we modelled age effects differently here and in the GWAS.

Detailed comments:

lines 50-51: The first question here ("What are the genomic patterns of homozygosity caused by inbreeding?") seems very specific compared to the much more general questions that follow. Also, at this stage of the manuscript the reader has no clue why it is important to know the genomic patterns of homozygosity. Please clarify.

R6: We rewrote the introduction to be a bit broader and hopefully clearer in outlining the questions and their importance.

lines 66-77: I found this introduction to runs of homozygosity a little superficial. There is no mention of the dependence of the length of ROH on the time to the next common ancestor, no mention of the effects of the demographic history of a population (except through a somewhat oblique reference to 'background relatedness'), and no mention of the effects of selection. Please provide some more background here.

R7: We have now written a section on ROH length, inbreeding and demographic history (line 74 ff.), and also briefly mention the idea that ROH length classes are differently enriched for deleterious alleles. However, for the reasons outlined in R2, we have kept these explanations relatively brief.

line 87: An additional study that probably appeared after the authors submitted this manuscript is Niskanen et al. 2020, PNAS (Consistent scaling of inbreeding depression in space and time in a house sparrow metapopulation).

R8: Thank you. We have added this study.

lines 111-112: Please specify in the Methods what 'complete annual survival and genomic data' implies. Does it imply that you only used animals that were found dead (as indicated in the line above) and from cohorts where everyone had died by the end of the study? And what settings were used to define complete genomic data?

R9: Complete annual survival data includes individuals for which both the birth year and the death year have been recorded, which we have now clarified in lines 429 ff.. Complete genomic data includes individuals which were genotyped on either of the

high density or low density SNP chips and passed the quality checks, as described in lines 441-452.

line 115: Remove one 'on'.

R10: Done.

line 134: How can the mean numbers of ROH in these two subsets (183 and 174) be lower than in the complete dataset (194)?

R11: Here, we only summarised the extremes of the distribution (top 1% on both sides of the inbred-outbred spectrum). I.e. the most inbred individuals have fewer ROH because they have on average much longer ROH. The most outbred individuals have fewer ROH because they are much less homozygous than the average individual. In response to another reviewer comment, we now summarise ROH only for the seven most inbred and outbred individuals to match the the individuals shown in Figure 1A.

line 138: Please explain (possibly in the Methods) why you used 500Kb windows.

R12: Though somewhat arbitrary, these windows are broadly comparable (given the relative SNP density) to similar studies (e.g. Kardos 2015, 2017). Each window contains on average 85 SNPs and averages noisy SNP variation well enough while still providing a relatively local genomic estimate. We now clarify this in lines 501 ff.

Figure 1B: Please provide and label the y axis of the density plot insert.

R13: Done.

lines 209-210: It did not become clear to me how you arrived at this conclusion. Please explain.

R14: In the GWAS analysis, we tested whether ROH status affects survival at every SNP position in the genome. As a thought experiment, let's assume that there are no recessive deleterious alleles at all causing inbreeding depression. In this case, we would expect that all ROH status effects (i.e. their model estimates or slopes) are essentially random. Because they are random, we would expect that ROH status effects on survival across SNPs would as often be positive as it would be negative, i.e. follow a 50/50 distribution. However, we find that ROH status has a negative effect on survival much more often than it has a positive effect for all 400K SNPs and models. While ROH effects at most of these SNPs are not "genome-wide significant", we can still infer something from the distribution: That ROH have (more often than expected by chance) negative effects because they reveal the effects of deleterious recessive alleles. However, the effect of those alleles is usually not strong enough (or they are not common enough) to be genome-wide significant. A conclusion from this is that many deleterious alleles are spread across the genome, and have (rather small) deleterious effects when expressed within ROH. Overall, this is a somewhat unusual way of

interpreting GWAS results and only possible because, unlike in an additive effects GWAS, we have an expected direction of the ROH effect. We elaborate a bit more on this in the paper now (e.g. lines 261 ff.).

lines 210-212: How long, on average, were the ROHs that these three SNPs were in and how do the lengths compare to the distribution of ROH lengths in the population?

R15: As explained in response R2, we think that the link between ROH length and deleterious alleles needs population genetic expectations and a different sample, which is why we would prefer not to go into these details within the current paper. We still checked: the mean ROH length is 3.1 Mb, and the mean ROH length at each of the five GWAS peaks from the new allele-based ROH GWAS (see R47) are: 6.8, 2.8, 5.4, 6.1 and 7.4 Mb. However, it is probably also important to take the ROH prevalence at each of these positions into account.

line 248: Large and small are relative terms. Hence, it would be good to say that here 'very large sample sizes' refers to more than 100'000 individuals.

R16: We clarified this.

lines 251-264: Taken at face value, this paragraph contradicts itself. The paragraph starts by pointing out how this and other genomic studies have revealed more inbreeding depression than previously thought. But it ends with the statement that this study revealed low to moderate inbreeding load. Many readers will likely find this contradictory. It would be important to clarify that genomic studies have found more inbreeding depression than studies of the same species that had used incomplete pedigrees. But that in comparison to other species the inbreeding load among Soay sheep is low to moderate. So, one is a technical reason the other a biological reason.

R17: This was slightly confusing, yes. We rewrote this section and hope it is clearer now. In general, we are not convinced that expressing inbreeding depression as lethal equivalents is still useful in the genomics era. This is because the number of lethal equivalents is simply a transformation from the slope of inbreeding in a mixed model (though a log model instead of a logit model). However, as correctly pointed out by Nietlisbach et al. (Nietlisbach, Muff, Reid, Whitlock, & Keller, 2019), lethal equivalents are not all equivalent, as these estimates depend on the fitness trait under consideration, the method of estimating inbreeding and other life-history and environmental variables. When estimating inbreeding depression based on a mixed model, all these variables are transparent, while the entity of a lethal equivalent in itself suggests comparability across studies, which is not the case. We compared our estimate to other estimates from appropriate statistical models, which are described in Nietlisbach et al.. However, studies vary in the depth and completeness of the pedigree or level of genomic information. It is likely that in some studies, inbreeding depression had to be much stronger to be detected (i.e. significant) based on a few microsatellites or incomplete pedigrees compared to inbreeding measured precisely

with genomic markers. The subset of studies we compared our estimate to (those from Nietlisbach et al., Table 1) is thus not unlikely to be an upwardly biased subset of inbreeding depression studies. We now added more explanations (lines 354 ff.) to make clear that estimates of lethal equivalents might well change in magnitude in the genomics era.

lines 261-264: This is true only if genetic load is considered the same as inbreeding load. Drift load would show a very different pattern. Since different authors define genetic load differently, it would be important to clearly define it here.

R18: Agreed, we changed this to inbreeding load.

line 270: Keller et al. 2008, Proc. Roy Soc. provide additional but mixed evidence for increased inbreeding depression later in life.

R19: Thanks for this relevant paper, which we have included now.

lines 292-293: I did not understand what was meant by 'particularly towards larger effect sizes'. Please clarify.

R20. Particularly in Figure 3B, it is apparent that the proportion of negative ROH in purple relative to positive ROH effects in yellow becomes higher for smaller p-values. For example, there are only a few more negative ROH effects which have p-values of 0.75, but many more negative ROH effects which have p-values of 0.01. The same is true for the model estimates in Figure 1A: Looking at a small estimate of 0.01, there are nearly as many positive and negative ROH effects. Looking at a large estimate of 0.5, there are many more negative ROH effects than positive ones. This fits to what we would expect when the ROH estimates really capture the effects of many mildly deleterious alleles. We would expect ROH effects to be negative, and we would expect them to have stronger effects and smaller p-values on average than the positive ROH effects (which are mostly due to chance).

line 338: Ref. 63 used a MAF cutoff of 0.01. Did you really use a cutoff of 0.001 here?

R21: Yes, we used a MAF cutoff of 0.001.

lines 341 and 345: The main text refers to 189 individuals genotyped with the high density SNP (line 116). Please clarify.

R22: There was a typo in line 341. We genotyped 189 sheep on the HD chip, but only 188 were retained after quality control. We state this clearly now in the genotyping methods section.

lines 376-378: How do these parameters compare to the recommendation of Meyermans et al. 2020, BMC Genomics?

R23: Meyermans et al. analysed medium density SNP chip data, mostly based on 50K chips. Hence, the tested SNP density is around ten times lower as in our study. These are datasets which are generally much more sensitive to changes in the PLINK parameters than high density SNP datasets. However, we did omit LD pruning and only applied MAF pruning with a very low threshold (0.001), which aligns with their recommendations. They further recommend to carefully adjust the PLINK kb/SNP minimum density setting, which is less relevant in our case due to high SNP density, similar to other parameters. We did also allow for the occasional heterozygote genotype in an ROH as recommended.

The main parameter influencing our inference (based on our trials) is the minimum ROH length, which we could have decreased below 1.2Mb and hence recovered more ROH. However, we decided to focus on longer ROH, as these result from more recent inbreeding and potentially provide greater power for inferences within a GWAS, and because there is little individual variation in these shorter ROH. The chosen threshold is also similar to what is used in human studies with similar SNP density (e.g. Clark et al., 2019)

lines 386-387: Please provide a reference.

R24: We now cite McQuillan et al. 2008.

line 393: I believe the 0.5% is based on the study by Pemberton et al. 2012? If this is correct, please cite this reference here.

R25: The 0.5% were chosen somewhat arbitrarily as the two extremes which fall outside 99% of the data. Pemberton et al. use the same threshold (Pemberton et al., 2012), and we now cite them in the Methods section.

line 421 and Supplementary Table 4: Please justify why you used a pedigree-derived additive relatedness matrix in the animal models rather than a genomic relatedness matrix. The latter would seem more natural given the use of ROH as a measure of inbreeding (see e.g. Luan et al. 2014, Genetics Selection Evolution).

R26: We decided on this simple solution for technical reasons. A GRM would be very large and computationally demanding and not straightforward to fit in INLA. Moreover, it is very unlikely that the additive genetic variance estimate would change our inbreeding depression results, because:

- 1) Additive genetic variance estimated from SNPs and pedigree has previously been shown to be very similar (Bérénos, Ellis, Pilkington, & Pemberton, 2014). This is because the Soay sheep pedigree is deep and relatively complete.**
- 2) Additive genetic variance in survival is very low (Supplementary Table 4), in line with previous results for fitness components in the study population (Johnston et al., 2013; Morrissey et al., 2012). We now explain this in the methods section (Lines 537 ff.)**

line 433: Please provide a reference for the choice of priors.

R27: There are no clear recommendations for priors in INLA, both due to its novelty and because these choices are very specific to the dataset and model. It has been shown that in a given range, estimates are quite insensitive to priors, but this is for inverse-gamma not for log-gamma as used in this study (Holand, Steinsland, Martino, & Jensen, 2013). We did an unsystematic comparison between INLA and other mixed modelling packages (MCMCglmm and lme4) and found that with this set of priors, all modelling package gave similar estimates for variance components.

line 477: I would prefer a quantitative statement here instead of 'negligible'. Supplementary Table 4 allows such a quantitative statement.

R28: We now state the estimated variance directly in the text.

line 498: Is an 'at' or 'by' missing in this sentence?

R29: Yes, thanks for spotting this.

Reference list: The citations need some attention. Journal names and abbreviations, the use of upper and lower case, and the use of page numbers are inconsistent. For some references information is missing (e.g. ref 23).

R30: We carefully corrected the reference list now.

Supplementary Figure 5: What are the error bars? 95% CIs?

R31: Yes, we now added this to the legend.

Supplementary Table 1: Please indicate in the caption what N(Snps) stands for.

R32: N(SNPs) is the number of SNPs on each chromosome, we added this now, thanks.

Reviewer #2 (Remarks to the Author):

Summary:

In this study, Stoffel et al. quantified the impact of inbreeding on annual survival in 5,952 wild Soay sheep. Their analyses included 1) identification of ROH in each individual based on imputed SNP genotypes at 417 thousand loci, 2) modeling the effect of Froh on annual survival in a Bayesian framework, and 3) a modified genome-wide association study to reveal the genetic architecture of inbreeding depression. They report evidence of inbreeding depression caused by many recessive deleterious mutations contained in large runs of

homozygosity. They find that annual survival is reduced by 68% in an individual with Froh 10% above the population-wide mean, and that this reduced survival primarily manifests in the first few years of life.

Evaluation:

This is an impressive study because it combines a comprehensive genomic analysis with detailed life history data in a large number of samples collected over a period of several decades. This is extremely difficult to do in a wild population, and such studies remain rare. With this sample size and density of genomic data, the results are quite convincing. The methods are robust and detailed sufficiently, and I was very pleased to see that all of the analysis code has been made accessible. The manuscript is clear, well-organized, and cites appropriate literature. The results are interesting and add to our understanding of the genomic basis of inbreeding depression. In my opinion, the manuscript is of sufficient quality and novelty to interest the diverse readership of Nature Communications. I have only minor suggestions for the authors, as described below.

Thank you very much!

Minor suggestions:

- A figure showing the distribution of minor allele frequencies would be a helpful addition to the supplement. A mean minor allele frequency of 23% is very high, though perhaps not unexpected here due to the fact that the population is small, inbred, isolated, founded from ~100 individuals in the 1930s, etc. These factors make the population rather unique, and probably influence the genetic architecture of inbreeding depression in this system (i.e. few large effect mutations, evidence of many weakly deleterious recessive loci across the genome), which may mean the results would not hold generally in other populations with different demographic histories.

R33: We added a new Supplementary Figure 3 showing the MAF distribution in the imputed dataset. We agree that the genetic architecture of inbreeding depression will vary depending on the demographic history of the species, which in turn impacts the distribution of deleterious alleles. We hoped to have expressed this clearly in the discussion (lines 338 ff.). Note that the MAF allele frequency distribution is also influenced by the fact that we are using SNPs selected for the arrays, for which high MAF across multiple sheep breeds was a criterion.

- It would be interesting to know whether Froh has changed over the course of the study period. Is there evidence of any change over the years or over the generations within the pedigree? If so, it would be interesting to note, and if not, it would provide more confidence in the results.

R34: This is an interesting question. We show the plot below and now included it as a new Supplementary Figure 5 to show that F_{ROH} has not changed across the study period, which is further confirmed by a simple linear model of F_{ROH} with birth year fitted as numeric predictor ($\beta_{birth\ year} = 0$, 95% CI [0,0], $p = 0.588$)

Rebuttal Figure 2: Inbreeding coefficients F_{ROH} across birth years.

- Showing the survival probability curves (as in Fig. 2C) for the other age classes (years 2, 3, 5, 6, 8+) in the supplement would aid transparency.

R35: We added a new Supplementary Figure 8A, which now shows survival curves for all age classes up to 9 years. In response to another reviewer comment, we also added a Figure 8B which shows the model predictions on the log-odds scale.

- 268: The relevance of mutation accumulation in this system is not clear to me. I am not sure that the gradual build up of new mutations over time in the Soay population due to long-term reduced inefficiency of selection is the right framework for understanding inbreeding depression within this population. The literature on mutation accumulation is certainly important, but the subsequent two paragraphs (282-310) about the distribution of dominance and deleterious mutation effect sizes seem far more relevant to understanding the nature of genetic load and inbreeding depression in this system.

R36: The paragraph on mutation accumulation discusses a different result than the following two paragraphs - that inbreeding depression becomes weaker in older individuals. However, it is not easy to think about what the expectation here would be. Mutation accumulation does provide an expectation: If more recessive mutations become deleterious in older individuals, inbreeding depression should increase in magnitude. There is both theoretical (Charlesworth & Hughes, 1996) and empirical work (Keller, Reid, & Arcese, 2008) on the connection between inbreeding depression and age under the assumption of mutation accumulation, which seemed to provide an appropriate framing for the discussion of these results. However, as we showed, the sample actually changes as inbred individuals die early, and this is potentially a much more important driver of the change in the magnitude in inbreeding depression, which actually becomes weaker with age in our population. We have retained and slightly rewritten the paragraph and hope the reviewer is ok with this decision, but we are happy to change it if there are serious doubts about our argumentation.

- 344: Was the same MAF filter used on the high density chip, which was run on far fewer individuals? MAF of .001 in a sample of 188 individuals is less than 1 allele.

R37: Thanks for noticing. In the HD sample, we retained all SNPs which were not monomorphic. We clarified this in the Methods now.

- 432-433: Should this be referring to the fixed effects, not the priors on the random effects?

R38: These are indeed the priors for the random effects, which are estimated in terms of precision rather than variance in INLA. This is a bit unusual and we now state this explicitly in the methods.

- Supplemental figure 7: A color-coded legend would be helpful here. The line colors are not explicitly mentioned in the caption.

R39: We have now added a color legend to the figure and added a more detailed description.

Reviewer #3 (Remarks to the Author):

This is a very interesting paper. It is certainly one of the most comprehensive, if not the most comprehensive, analysis of inbreeding depression in a wild population to date. The paper looks at many of the key components of inbreeding and inbreeding depression: effects of inbreeding on a key fitness component, lethal equivalents, the genomic distribution of ROH density, and it identifies genomic regions that likely contribute to inbreeding depression using association analysis and a purely genomic approach. The methods section is comprehensive, and the interpretation of results is appropriate.

I strongly support publication of this paper in any journal. However, I suggest addressing some important issues in the paper beforehand. My comments are below. The most important are marked with an asterisk. In particular, I suggest digging deeper into the mechanisms leading to the high variance in the distribution of ROH density across the genome, and considering a modification to the association analysis to increase statistical power and the accuracy of estimated locus-specific effect sizes on fitness.

R40: We really appreciated the suggestions and included a complete new set of analyses and (updated) figures, both for the relationship between recombination and ROH and for the GWAS, as described below.

68-70: It would be more precise and succinct to say that short ROH originate from distant ancestors (i.e., common relatives of the parents many generations ago).

R41: We agree and changed this.

131: What is the motivation for using 1.2 Mb as a minimum length threshold?

R42: Please also see response R49. 1.2 Mb is chosen somewhat arbitrary, but the initial idea was to define 1.2Mb as the expected mean ROH length when the underlying haplotypes had a common ancestor 32 generations ago (see Methods and now Figure 1B). That said, the variance around this average will be large (as you mention below). However, while the threshold is somewhat arbitrary, we decided not to call very short ROH, as these are less likely to be different to homozygosity caused by LD (see Supplementary Figure 11 for a plot of the LD decay), and as such potentially less informative in the ROH GWAS analysis. We also describe this briefly in lines 484 ff.

132-134: I find the wording awkward and had to read this a few times to figure out the meaning. You could instead write something like "the seven most inbred sheep had longer ROH (mean: 5.27 Mb vs. 2.76 Mb), and slightly more abundant ROH (mean: 183 vs. 174) than the seven least inbred sheep"

R43: It seems like not only the wording was a bit confusing here. The 1% most at least inbred sheep actually referred to many more individuals (120) than shown in the plot. However, we simplified this now by referring to only the seven most inbred and outbred sheep which are also shown in the plot, and we also re-worded the sentence to make it clearer.

139-146: I wonder if the regions with exceptionally few ROH occur in regions that also have high admixture from Dunface. Do you have a map of the abundance of Dunface haplotypes across the Soay genome?

R44: This is a very interesting possibility. We do not have such a map. We see a number of complexities to this analysis including that the Dunface is extinct and we do not currently have high density SNP genotypes for the other breed with Dunface admixture, the Boreray. We feel this analysis is beyond the scope of this particular MS at this time.

*142-146 and elsewhere relating to ROH abundance and recombination rate: This is very interesting and important, and the field would benefit from digging more deeply into the reasons for this pattern. The relationship between ROH density and recombination rate is substantially more complicated than it appears by reading the relevant literature. First, it seems worth putting a figure showing something like that in Fig S4 in the main text - for example, a single plot showing the data from all chromosomes combined at once. Then I suggest adding some material on the potential mechanisms underlying this pattern.

R45: Thanks, we appreciate this and added a new figure and new analyses as described below.

The main finding here is more ROH in regions with lower recombination rate. In lines 237-239, the potential influence of recombination rate, selection, and gene density are rightly pointed out. I agree with the proposed potential selective explanations: ROH deserts are good candidates for harbouring strongly deleterious recessive alleles or loci under balancing selection; ROH islands may well be regions where selective sweeps have

occurred. This analysis and argument would be strengthened by seeing if there are any ROH deserts in regions of the genome without exceptionally high recombination rate. In such cases, it would be hard to point to something related to recombination rate (see next paragraph) as the most likely explanation. However, the ROH islands and deserts might all be located in regions represented in the far left and right ends of the x-axes in Figure S4, respectively, in which case it is more difficult to point to selection as the most likely explanation. This is particularly important when ROH density is clearly associated with recombination rate (this was not the case in the cited Scandinavian wolf paper on ROH). I suggest investigating this, albeit briefly, at least as presented in the main text.

An issue with interpretation of ROH results that is particularly prevalent in the human genetics literature, and relevant here I believe, is that recombination by itself (i.e., without an additional influence of selection) cannot affect the TRUE abundance of ROH. Regions with higher recombination rate produce shorter ROH than parts of the genome with low recombination rate. This means it is generally easier to detect ROH in regions with low recombination because such ROH are more 'obvious' to ROH detection algorithms. This can by itself create the appearance of a correlation between ROH density and recombination rate. We looked at this in detail in our paper on genomic analysis of inbreeding in flycatchers (Kardos et al. 2017, Genetics). See in particular Figure 7 in that paper, which shows that the density of DETECTED ROH can vary with recombination rate across genomes where inbreeding (heterozygosity) is really held constant across the genome. The issue is that ROH with a

given coalescent time will be tiny in regions with very high recombination rate, and huge in regions with very low recombination rate. This phenomenon likely explains at least part of the genome-wide variation in ROH density here... You can easily detect big ROH in regions with low recombination rate, whereas it is harder to detect them in regions with high recombination rate because they are so short.

My suggestion is to at least raise this in the Discussion. Even better would be to test whether this is likely to explain the results by seeing if ROH density is correlated with heterozygosity (proportion of het SNPs in the same windows as ROH density is analyzed); if this association is significant, it would suggest that ROH abundance is indeed tracking variation in true total IBD across the genome. One caveat is that ascertainment bias associated with the construction of the SNP chips might make this difficult (genome sequence data would be better), but I am not sure. Another caveat is that with 'only' < 500K SNPs, one cannot account for the contribution of very short ROH.

R46: We now investigate this in much more detail. First, we added an additional Figure (Figure 2), showing the relationship between ROH density, recombination rate and SNP heterozygosity, both for the full data (all chromosomes) and only for ROH islands and deserts. To disentangle whether variation in ROH density is mostly shaped by variation in recombination rate or indeed tracks the underlying IBD, we modelled ROH density as a function of recombination rate and SNP heterozygosity (lines 166 ff.), all of which were calculated as 500Kb window averages, while also fitting a chromosome identifier as random effect.

To compare the relative contributions of heterozygosity and recombination rate to variation in ROH density, we calculated their semi-partial R^2 . Using the full data, recombination rate only has a small impact on genome-wide ROH density (semi-partial $R^2 = 0.04$, 95% CI [0.02, 0.06]), while the variance explained by heterozygosity is quite large (semi-partial $R^2 = 0.38$ 95% CI [0.36, 0.40]). The same pattern holds when modelling only regions designated as ROH islands and deserts where heterozygosity explains a substantial part of their ROH abundance (semi-partial $R^2 = 0.89$, 95% CI [0.82, 0.93]) while recombination rate explains a small part of the variation (semi-partial $R^2 = 0.07$, 95% CI [0.004, 0.114]). Moreover, Figure 2C shows that some of the ROH islands and deserts indeed have very similar recombination rates.

*195-216: This is a very nice analysis and I am really glad to see it done. I particularly appreciate that it controls for genome-wide identity disequilibrium (469-470). However, there might be a missed opportunity to identify more loci contributing to inbreeding depression, and to precisely estimate their fitness effects. Consider a diallelic locus where allele A has a strong deleterious, mostly recessive effect on fitness, and allele B causes no inbreeding depression. The analysis here assumes that ROH presence/absence affects fitness equally regardless of allele/haplotype identity. This means that the fitness effects of homozygosity for the two alleles are essentially averaged in the analysis, if I understand the model correctly. In our hypothetical example, the present analysis would have perhaps substantially underestimated the effect size. Treating ROH status for each allele separately (i.e., test for an effect of ROH presence/absence separately for each allele) could in principle greatly increase power, and provide a much better estimate of locus/allele-specific fitness effect. An alternative, if the data are phased, would be to conduct the association test on haplotype-specific ROH status.

R47: Yes, it is true that ROH effects of both alleles were averaged in our initial GWAS analysis by only fitting one ROH status effect, which had the advantage of a larger sample size (more ROH in the ROH status group) for a given test, and only half as many statistical tests overall.

However, we agree that testing ROH status per allele is more concise and potentially more powerful and therefore have changed our analysis now. We repeated the GWAS and included two binary predictors, one for ROH status for allele A and one for ROH status for allele B. To account for double as many statistical tests, we doubled the number of effective tests for the Bonferroni correction and got a new genome-wide significance threshold of $p < 6.38 * 10^{-7}$ (see line 616). Reassuringly, the ROH status (for one of the alleles) at the same three loci as before (chromosomes 3, 10, 21) reached genome-wide significance. The new GWAS had indeed more statistical power, as it also uncovered two further regions with putative strongly deleterious alleles on chromosomes 14 and 18 (Figure 3C, Supplementary Table 5). Moreover, and somewhat surprisingly, ROH of specific haplotypes at two further loci increased survival probabilities. At a first glance, these regions are very interesting but a deeper analysis is not related to inbreeding depression and thus outside the scope of this manuscript. We therefore postponed this analysis for now and only talk about these positive associations between ROH and fitness briefly in the manuscript.

233-234: This figure would inevitably be considerably higher if you could detect shorter ROH (e.g., with whole genome resequencing). It would be worth mentioning this in the Discussion.

R48: We now mention this in the first paragraph of the discussion.

382-385: What is the reason for the threshold of 32 generations to the common ancestor haplotype? Measuring inbreeding due only to recent ancestors is much better done by defining the length threshold in terms of cM rather than Mb, specifically because the recombination rate obviously varies a lot across the genome. The way the analysis was done means that ROH of the same physical length (say 1.2 Mb) have very different average coalescent times in different parts of the genome; the time horizon therefore is quite 'fuzzy' in the analyses. I don't think this is a big problem. You could either redo the analysis using a cM-based threshold, or acknowledge that the realized g (line 384) must have a much higher variance across detected ROH of a given length class than would be the case with a cM-defined threshold.

R49: The choice is slightly (though not completely) arbitrary. We wanted to call ROH as short as possible, while trying to avoid calling ROH which are too short and mainly due to LD in the population (as these might be less informative for the GWAS). Based on the LD decay (Supplementary Figure 7), and our marker density, the smallest called ROH should probably be > 1Mb. In a first draft of the manuscript, we included a plot on the ROH length distribution (which we now included again in response to reviewer 1, see Figure 1B), which plots ROH length classes originating from exponentially increasing most recent common ancestors, with the smallest length class reflecting ROH with MRCA ~32 generations back in time.

We appreciate that physical length ROH distributions will have a larger variance than genetic map length ROH and mention this now in lines 316 and 495. We decided to call ROH based on physical length, as the imputation greatly increases the marker density and hence the power to call ROH based on physical positions accurately. The Soay sheep linkage map is based on ~39K SNP markers, which would decrease the marker density to infer ROH substantially. As ROH length considerations play only a minor part in this study, we felt that calling ROH based on the physical map positions and high marker density was the way to go here.

483-484: I suggest reporting the effective number of tests.

R50: We now report the effective number of tests in the methods section.

488: If the data will be embargoed then I suggest stating that here.

R51: We now plan to embargo the data for a year and state this explicitly in the manuscript now.

163 and Figure 2b: The point estimate is reported as a 68% reduction in the text, but in the

report of model estimates (in the parentheses), $1 - 0.68 = 0.32$ and its CI is given. Then the figure legend combined with Figure 2b suggests that the reduction in survival odds associated with a 0.1 increase in Froh is ~ 0.32 . First, I think the meaning of the parameter being estimated should be stated much more clearly. Additionally, the parameter estimate should be stated consistently everywhere it is mentioned (not as 0.68, and then $1 - 0.68$ elsewhere). It is hard to follow as written, and this would likely lead to some misinterpretation among readers. It is not clear to me which is correct.

R52: Yes, this is indeed a bit confusing. The model estimate (0.32) is the expected multiplicative change in the odds of survival for a 10% increase in Froh. This is expressed a bit simpler as a 68% reduction in the odds of survival. To avoid confusion, we now removed the CI in the abstract and clarified the relationship between the two numbers in the results paragraph.

263: Should this not be 'inbreeding load' instead of 'genetic load'.

R53: Yes and changed.

(signed)

Marty Kardos

Literature

- Bérénos, C., Ellis, P. A., Pilkington, J. G., & Pemberton, J. M. (2014). Estimating quantitative genetic parameters in wild populations: a comparison of pedigree and genomic approaches. *Molecular Ecology*, *23*(14), 3434–3451.
- Chakrabarty, A., van Kronenberg, P., Toliopoulos, N., & Schielzeth, H. (2019). Direct and indirect genetic effects on reproductive investment in a grasshopper. *Journal of Evolutionary Biology*, *32*(4), 331–342.
- Charlesworth, B., & Hughes, K. A. (1996). Age-specific inbreeding depression and components of genetic variance in relation to the evolution of senescence. *Proceedings of the National Academy of Sciences*, *93*(12), 6140–6145.
- Clark, D. W., Okada, Y., Moore, K. H. S., Mason, D., Pirastu, N., Gandin, I., ... Wilson, J. F. (2019). Associations of autozygosity with a broad range of human phenotypes. *Nature Communications*, *10*(1), 1–17. doi: 10.1038/s41467-019-12283-6
- Holand, A. M., Steinsland, I., Martino, S., & Jensen, H. (2013). Animal models and integrated nested Laplace approximations. *G3: Genes, Genomes, Genetics*, *3*(8), 1241–1251.
- Johnston, S. E., Gratten, J., Berenos, C., Pilkington, J. G., Clutton-Brock, T. H., Pemberton, J. M., & Slate, J. (2013). Life history trade-offs at a single locus maintain sexually selected genetic variation. *Nature*, *502*(7469), 93.
- Keller, L., Reid, J., & Arcese, P. (2008). Testing evolutionary models of senescence in a natural population: age and inbreeding effects on fitness components in song sparrows. *Proceedings of the Royal Society B: Biological Sciences*, *275*(1635), 597–604. doi: 10.1098/rspb.2007.0961
- Morrissey, M. B., Parker, D. J., Korsten, P., Pemberton, J. M., Kruuk, L. E., & Wilson, A. J. (2012). The prediction of adaptive evolution: empirical application of the secondary theorem of selection and comparison to the breeder's equation. *Evolution: International Journal of Organic Evolution*, *66*(8), 2399–2410.
- Nietlisbach, P., Muff, S., Reid, J. M., Whitlock, M. C., & Keller, L. F. (2019). Nonequivalent lethal equivalents: Models and inbreeding metrics for unbiased estimation of inbreeding load. *Evolutionary Applications*, *12*(2), 266–279.
- Pemberton, T. J., Absher, D., Feldman, M. W., Myers, R. M., Rosenberg, N. A., & Li, J. Z. (2012). Genomic patterns of homozygosity in worldwide human populations. *The American Journal of Human Genetics*, *91*(2), 275–292.
- Schielzeth, H. (2010). Simple means to improve the interpretability of regression coefficients. *Methods in Ecology and Evolution*. Retrieved from <http://onlinelibrary.wiley.com/doi/10.1111/j.2041-210X.2010.00012.x/pdf>
- Szpiech, Z. A., Xu, J., Pemberton, T. J., Peng, W., Zöllner, S., Rosenberg, N. A., & Li, J. Z. (2013). Long runs of homozygosity are enriched for deleterious variation. *The American Journal of Human Genetics*, *93*(1), 90–102.

Zhang, Q., Guldbrandtsen, B., Bosse, M., Lund, M. S., & Sahana, G. (2015). Runs of homozygosity and distribution of functional variants in the cattle genome. *BMC Genomics*, *16*(1), 542. doi: 10.1186/s12864-015-1715-x

REVIEWER COMMENTS

Reviewer #1 (Remarks to the Author):

Genetic architecture and lifetime dynamics of inbreeding depression in a wild mammal

I thank the authors for the detailed explanations that accompany the revision. Most of my questions and concerns have been answered. However, one major point remains: the specification and interpretation of the Bayesian animal model as presented in the text, Figure 3, Supplementary Table 4, and Supplementary Figures 8 and 9.

The authors argue in the rebuttal letter that the difficulties involved in the model interpretation stem from the non-linear relationship between the log-odds estimates of the logistic regression model and the survival probability (response R4). This is certainly part of the problem, and I will argue below that too much of the interpretation is on the log-odds scale rather than on the survival probability scale. It is the latter that matters biologically. However, I think there is also an issue with model fit.

Two statistical issues are still unresolved: 1) the interpretation of the model results in the main text, and 2) the fit of the model to the data seems poor. I will explain both points in detail below.

1) The model results are accurately represented in Figure 3 but misrepresented in the text. As shown in Figure 3C, the model results imply that lambs experience more not less inbreeding depression in survival than older sheep, at least up to a FROH of 0.4 (very few individuals have higher FROH). This is also what the model results in Supplementary Table 4 state. If one puts all of the parameter estimates from Supplementary Table 4 into the model equation given in the Methods (lines 525 and 526), while holding sex and twin constant at 1 (male) and 0 (no twin) as in Figure 3C, one arrives at the following estimates of survival for animals with average FROH (0.24) and animals with an FROH of 0.34:

lambs: 0.51 and 0.25
age=1: 0.96 and 0.84
age=4: 0.93 and 0.83
age=7: 0.88 and 0.81

(The R code for these estimates is at the end).

Thus, according to the model, inbreeding depression in lamb survival is far more pronounced than at the other ages. As they should, these values correspond with what is shown in Figure 3C. Thus, Figure 3C is an accurate description of the model results. What is off is the interpretation in the main text. Contrary to the claims on lines 207 ff., the model results suggest that inbreeding depression in lambs is more pronounced than in older sheep. Also, contrary to the claims in the text, the model does not give strong indications for a decline of inbreeding depression in adults, except perhaps in the very oldest age classes (e.g. age=7 above shows only an 8% decline in survival, compared to 11 and 13% in 4- and 1-year-olds).

Since being a lamb is part of both age-related covariates, LAMB and AGE, it is impossible to say whether there is evidence for declining inbreeding depression in adults. The apparent decline could be exclusively due to the decline of inbreeding depression from lambs to adulthood.

In summary, the model results as given in Supplementary Table 4 do not support the conclusions drawn in the main text. This discrepancy stems from the fact that the authors base much of their interpretation in the text on the log-odds scale rather than the biologically meaningful survival scale. However, as I will discuss next, there are good reasons to think that the model used to estimate the parameters in Supplementary Table 4 does not fit the data well and that the parameter estimates therefore are off.

2) Supplementary Figure 9 shows the age-specific slope of FROH (β_{FROH}) on the log-odds scale when estimated in separate models for each age class. These estimates from separate models for each age class can be compared to the same slopes predicted from the model used in the manuscript. The R code at the end calculates these slopes from the parameter estimates given in Supplementary Table 4 and plots them for each age class. A comparison of this plot with Suppl. Figure 9 suggests that the model implemented in the manuscript does not fit the data well. The model implemented in the manuscript clearly overestimates the difference of β_{FROH} between lambs and one-year-olds, and Suppl. Figure 9 does not show the linear increase in β_{FROH} from age 1 to age 9 predicted by the model in the manuscript.

All in all, this suggests that the model as implemented in the manuscript does not fit the data well and that the resulting parameter estimates are likely off.

Point 1) can be fixed by using the survival probability scale for the interpretation of the model output. It is more difficult for me to suggest how to fix point 2) given my ignorance of the raw data. However, given Supplementary Figure 9 I would suggest that a model with categorical age effects and the corresponding interactions with FROH might fit the data much better. This specification of the age effects makes the model a bit more parameter-rich, but given the large sample sizes the parameters should be well estimated.

Detailed comments:

- lines 204-206: the odds ratio of 0.32 does not represent the average effect on St. Kilda of a 10% increase in FROH. Instead, 0.32 is the odds ratio of a 0.1 increase in FROH in female, non-twins of age 3.5 years. It is not the average inbreeding depression experienced in the population. This should be made clear.

- lines 349-354: The odds ratio from this study is not comparable to estimates from other studies that are not based on odds ratios. In addition, as shown in point 1) above, odds ratios can be misleading on

the survival scale: Note the poor correlation between odds ratios and inbreeding effects on the survival scale in this data set (R code below). It would be better to base this comparison on lethal equivalents.

R Code:

=====

```
library(boot)
```

```
# average age according to Suppl. Figure 9
```

```
av_age <- (2134*1 + 1589*2 + 1282*3 + 1100*4 + 910*5 + 720*6 + 554*7 + 422*8 + 254*9)/(2134 + 1589 + 1282 + 1100 + 910 + 720 + 554 + 422 + 254)
```

```
# parameter estimates in linear predictor according to Suppl. Table 4 for males (sex=1) and not twins (twin=0)
```

```
int <- 3.34
```

```
Froh <- -1.14
```

```
Age <- -0.21
```

```
Lamb <- -3.41
```

```
Sex <- -0.63
```

```
fa_int <- 0.17
```

```
fl_int <- 0.62
```

```
# calculate linear predictor based on the parameter estimates above
```

```
# linear predictor for lamb (age=0) at average F_ROH (=0)
```

```
linpred_L_0 <- int + Froh*0 + Age*(0-av_age) + Lamb + Sex*1 + fa_int*(0-av_age)*0 + fl_int*0*1
```

```
surv_L_0 <- inv.logit(linpred_L_0)
```

```
surv_L_0
```

```
# linear predictor for lamb (age=0) at F_ROH of 0.34 (=1)
```

```
linpred_L_1 <- int + Froh*1 + Age*(0-av_age) + Lamb + Sex*1 + fa_int*(0-av_age)*1 + fl_int*1*1
```

```
surv_L_1 <- inv.logit(linpred_L_1)
```

```
surv_L_1
```

```
# inbreeding depression lambs
```

```
ID_Lamb <- (surv_L_0 - surv_L_1)/surv_L_0
```

```
# odds ratio for inbreeding effect in lambs
```

```
OR_Lamb <- (surv_L_1/(1-surv_L_1))/(surv_L_0/(1-surv_L_0))
```

```
# linear predictor for age=1 at average F_ROH (=0)
```

```
linpred_1_0 <- int + Froh*0 + Age*(1-av_age) + Sex*1 + fa_int*(1-av_age)*0
surv_1_0 <- inv.logit(linpred_1_0)
surv_1_0
```

```
# linear predictor for age=1 at F_ROH of 0.34 (=1)
linpred_1_1 <- int + Froh*1 + Age*(1-av_age) + Sex*1 + fa_int*(1-av_age)*1
surv_1_1 <- inv.logit(linpred_1_1)
surv_1_1
```

```
# inbreeding depression age=1
ID_1 <- (surv_1_0 - surv_1_1)/surv_1_0
# odds ratio for inbreeding effect in sheep age=1
OR_1 <- (surv_1_1/(1-surv_1_1))/(surv_1_0/(1-surv_1_0))
```

```
# linear predictor for age=4 at average F_ROH (=0)
linpred_4_0 <- int + Froh*0 + Age*(4-av_age) + Sex*1 + fa_int*(4-av_age)*0
surv_4_0 <- inv.logit(linpred_4_0)
surv_4_0
```

```
# linear predictor for age=4 at F_ROH of 0.34 (=1)
linpred_4_1 <- int + Froh*1 + Age*(4-av_age) + Sex*1 + fa_int*(4-av_age)*1
surv_4_1 <- inv.logit(linpred_4_1)
surv_4_1
```

```
# inbreeding depression age=4
ID_4 <- (surv_4_0 - surv_4_1)/surv_4_0
# odds ratio for inbreeding effect in sheep age=4
OR_4 <- (surv_4_1/(1-surv_4_1))/(surv_4_0/(1-surv_4_0))
```

```
# linear predictor for age=7 at average F_ROH (=0)
linpred_7_0 <- int + Froh*0 + Age*(7-av_age) + Sex*1 + fa_int*(7-av_age)*0
linpred_7_0
surv_7_0 <- inv.logit(linpred_7_0)
surv_7_0
```

```
# linear predictor for age=7 at F_ROH of 0.34 (=1)
linpred_7_1 <- int + Froh*1 + Age*(7-av_age) + Sex*1 + fa_int*(7-av_age)*1
linpred_7_1
surv_7_1 <- inv.logit(linpred_7_1)
surv_7_1
```

```
# inbreeding depression age=7
```

```

ID_7 <- (surv_7_0 - surv_7_1)/surv_7_0
# odds ratio for inbreeding effect in sheep age=2
OR_7 <- (surv_7_1/(1-surv_7_1))/(surv_7_0/(1-surv_7_0))

# compare inbreeding depression and odds ratios
ID_Lamb
OR_Lamb
ID_1
OR_1
ID_4
OR_4
ID_7
OR_7

# Estimate age-specific beta_FROH (as in Suppl. Fig. 9) and plot them

beta_FROH <- rep(NA,10)

beta_FROH[1] <- Froh + fl_int + fa_int*(0-av_age)
beta_FROH[2] <- Froh + fa_int*(1-av_age)
beta_FROH[3] <- Froh + fa_int*(2-av_age)
beta_FROH[4] <- Froh + fa_int*(3-av_age)
beta_FROH[5] <- Froh + fa_int*(4-av_age)
beta_FROH[6] <- Froh + fa_int*(5-av_age)
beta_FROH[7] <- Froh + fa_int*(6-av_age)
beta_FROH[8] <- Froh + fa_int*(7-av_age)
beta_FROH[9] <- Froh + fa_int*(8-av_age)
beta_FROH[10] <- Froh + fa_int*(9-av_age)

plot(0:9,beta_FROH,xlab='age')

```

Reviewer #2 (Remarks to the Author):

Evaluation

I have read the revised manuscript from Stoffel et al. and find that this updated version is actually weaker than the original. Unfortunately, changes to the original manuscript have introduced a number of serious problems (see #1, 2, 5 below). Mostly, the manuscript is weakened by the attempt to link patterns of ROH density to selection. Upon deeper reflection, a couple of issues that were present in the original version have also occurred to me and now seem like bigger liabilities in light of the other

changes (see #3, 4 below). My original assessment of the manuscript was that it represented a generally well-executed study of an important problem using an impressive dataset, constituting a genuine advance for the field. I still feel that the study is important, but the revised manuscript contains critical flaws that must be corrected.

Major Issues

1. Unsupported claims about the role of selection in shaping patterns of ROH

Throughout the manuscript (lines 31, 67-72, 166-168, 326-346), the authors state that ROH density could be driven by selection, and they claim that their results support this hypothesis. Other hypotheses, apart from varying recombination rate, are not mentioned. I was especially dismayed to see that the abstract was rewritten to remove relevant details about the study, and a statement attributing patterns of ROH density to selection was added. The specific claim here is that there is selection either for or against ROH in certain parts of the genome, leading to regions with particularly high or low ROH density. In fact, there is no formal assessment of whether the ROH density could be generated under neutrality. The authors acknowledge that local recombination rate could affect power to detect long ROH, thus impacting ROH density estimation. However, varying recombination rate is not the only confounding factor, and it may not even be the most important factor (see more on the recombination rate analysis in point 2 below).

In the absence of selection for/against ROH, the density (probability) of ROH should partially reflect variable N_e across the genome, which is generated through the interaction of demographic history (drift, inbreeding, admixture), mutation rate, prior history of selection, and recombination rate. This is assuming technical artifacts are not artificially inflating diversity in some regions. Perhaps some regions really do have more or fewer ROH than expected under neutrality, but this must be tested explicitly before making claims about selection. The proper null model must control for the inherent variability in ROH likelihood across the genome. Specifically, highly variable regions of the genome would be less likely to appear in ROH, not because the probability of IBD is different here, but because the probability of inheriting two copies of the same haplotype by chance (IBS) is lower than in other regions with low diversity. It is not sufficient to simply quantify ROH density and then presume that the variability is driven by selection on ROH. There will always be a top 0.5% and a bottom 0.5% of ROH density. The question is whether these are the products of selection, and that has not been shown. Maybe the observed distribution is exactly what you would expect, just by the demographic process and whatever amount of variation there was to start.

At the very least, the authors should look at levels of diversity within versus between individuals (i.e. heterozygosity versus π). If within-individual diversity is lower/higher than expected, conditional on between-individual diversity and the expected homozygosity due to inbreeding, then this could point to selection. The idea is that detecting selection requires controlling for both inbreeding and random sampling. The claims about selection governing ROH density must be removed or qualified, or there needs to be a better analysis to provide evidence of selection.

2. Questionable interpretation of recombination rate analysis

The purpose of the recombination rate analysis (lines 166-188) is unclear. The first sentence of this section seems to imply that the purpose might be to control for recombination rate in order to strengthen the claim that ROH density outliers are driven by selection. However, maybe this analysis is just meant to reassure us that there is sufficient power to detect ROH, and recombination rate itself is not the primary driver of ROH density (although, as the authors note, high recombination rate itself is not expected to affect the true ROH density).

I don't think there's a real issue of power here. Power to detect ROH is mostly a problem when trying to identify short ROH (tens to hundreds of KB), not long ROH as in this study. But showing that there's only a mild effect of recombination rate on ROH density is good for covering the bases, and the results are consistent with what has been seen before (e.g. Pemberton et al. 2012, Figure 8B,C), despite this study using physical rather than genetic lengths. To summarize the weak correlation of recombination rate with ROH density, a brief mention in the text and a supplemental figure or two (as in the original manuscript) would suffice.

The introductory and closing sentences of this paragraph, however, and lines 331-333 of the discussion, make me question whether there was a different intended message. The paragraph ends with "ROH density mostly reflects the underlying patterns homozygosity along the genome." Isn't this just saying runs of homozygosity reflect homozygosity? Seems obvious that regions with high heterozygosity by definition must not have a lot of ROH. All else being equal, we would not expect recombination rate variation to drive ROH density (of long ROH), only the expected lengths of ROH, and even then, it would probably have little effect on ROH caused by recent inbreeding anyway. I can imagine a scenario where regions of high recombination rate might also harbor elevated diversity, which could in turn influence ROH prevalence, but this would need to be shown. I recommend reducing the recombination rate analysis to a brief note, or at least reframing to remove the implication that it shows evidence for selection on ROH.

3. Lack of detail about demographic history in the main text

The original manuscript largely focused on the relationship between F_{roh} and inbreeding depression. Despite the lack of detail about the system in the introduction of that version, I felt that the results were pretty straightforward, and the details were there in the discussion and methods. In the revised manuscript, there is a lot more attention on the mechanisms underlying patterns of ROH, and the GWAS results/interpretations are a little bit different. In light of these changes, it has become necessary to provide the history of the study system in detail in the introduction. Our expectations about patterns of ROH and the effect sizes of deleterious mutations (see point 4 also) are impacted by the fact that this population has undergone multiple bottlenecks, experienced admixture, and has unusually high levels of

recent inbreeding. Also, the authors state that Soay sheep have been around for "thousands" of years. This is too vague. It is important to give proper context by disclosing these relevant details in the introduction. Lines 107-111 and 417-423 could simply be moved.

4. Conclusions about effect sizes and number of lethal equivalents fail to properly take demographic history into account

The authors find that inbreeding depression in this system is caused by deleterious alleles across many loci, with only a handful of mutations with large effect sizes. Further, the authors state that the number of lethal equivalents in Soay sheep is not particularly high, just moderate, and that perhaps this is due to better estimation from genomic data (lines 358-363). The major issue with both of these conclusions (small effect sizes, few lethal equivalents) is the failure to discuss the possibility that strongly deleterious mutations have been purged through repeated bottlenecks and long-term small N_e . This population has experienced very high levels of inbreeding, multiple bottlenecks (relocation to St. Kilda, colonization of Soay, initial domestication of sheep), and admixture. All of these factors likely shaped the load of segregating deleterious mutations. It seems reasonable that with the intense inbreeding and drift, most of the remaining recessive deleterious alleles would be relatively weak, as large effect mutations would have already been eliminated by selection, or were preferentially lost through repeated bottlenecks that would be less likely to preserve low frequency alleles. It therefore makes sense that few loci of large effect were discovered here, and that the number of lethal equivalents is not especially high. This needs to be discussed in the text, in part to ground the results, but also to highlight that this is a unique system with a particular history, and the findings are probably not universal.

5. Data embargo is unacceptable

The new data availability statement indicates the data will be under embargo for one year, with no justification. For a study of potentially high significance that will be published in a widely-read and open access journal, the data must be made available with the paper. I cannot think of a reason why this dataset should be kept out of the public domain. This embargo is completely at odds with the spirit of an open access journal.

Reviewer #3 (Remarks to the Author):

I am impressed by the revision and have only a few remaining minor comments below.

138: insert 'more than' before 'twice'.

142 (and other relevant text in the MS): I believe this actually refers to the largest fraction of the IBD part of the genome, not the largest fraction of ROH. It is not expected that ROH this long would outnumber shorter ones, as the distribution of ROH lengths is always expected to be exponential (i.e., many, many very short ROH and fewer long ROH), even if most of the genome is covered in very long ROH.

Figure 4: You might consider putting spaces in between each chromosome, and making the points smaller to increase the clarity of the P-value distribution across each chromosome. The P-values on the right hand side of each chromosome are covered up by those on the left hand side of the following chromosome.

Marty Kardos

Response to reviewers

Reviewer #1 (Remarks to the Author):

Genetic architecture and lifetime dynamics of inbreeding depression in a wild mammal

I thank the authors for the detailed explanations that accompany the revision. Most of my questions and concerns have been answered. However, one major point remains: the specification and interpretation of the Bayesian animal model as presented in the text, Figure 3, Supplementary Table 4, and Supplementary Figures 8 and 9.

R1: We are glad that the reviewer found our revised manuscript acceptable and appreciate the discussion around the modelling. Please see below for our responses.

The authors argue in the rebuttal letter that the difficulties involved in the model interpretation stem from the non-linear relationship between the log-odds estimates of the logistic regression model and the survival probability (response R4). This is certainly part of the problem, and I will argue below that too much of the interpretation is on the log-odds scale rather than on the survival probability scale. It is the latter that matters biologically. However, I think there is also an issue with model fit.

R2: We have thought about this in more detail now and agree with the reviewer. As we explain below, we have now changed our interpretation of the modelling estimates with a focus on the probability scale.

Two statistical issues are still unresolved: 1) the interpretation of the model results in the main text, and 2) the fit of the model to the data seems poor. I will explain both points in detail below.

1) The model results are accurately represented in Figure 3 but misrepresented in the text. As shown in Figure 3C, the model results imply that lambs experience more not less inbreeding depression in survival than older sheep, at least up to a FROH of 0.4 (very few individuals have higher FROH). This is also what the model results in Supplementary Table 4 state. If one puts all of the parameter estimates from Supplementary Table 4 into the model equation given in the Methods (lines 525 and 526), while holding sex and twin constant at 1 (male) and 0 (no twin) as in Figure 3C, one arrives at the following estimates of survival for animals with average FROH (0.24) and animals with an FROH of 0.34:

lambs: 0.51 and 0.25
age=1: 0.96 and 0.84
age=4: 0.93 and 0.83
age=7: 0.88 and 0.81

(The R code for these estimates is at the end).

Thus, according to the model, inbreeding depression in lamb survival is far more pronounced than at the other ages. As they should, these values correspond with what is shown in Figure 3C. Thus, Figure 3C is an accurate description of the model results. What is off is the interpretation in the main text. Contrary to the claims on lines 207 ff., the model results suggest that inbreeding depression in lambs is more pronounced than in older sheep.

R3: We agree that this is correct if the sole focus was on individuals with inbreeding coefficients in the range between $F_{roh} = 0.24$ and $F_{roh} = 0.34$. However, we would like to argue here that strongly inbred individuals do have F_{roh} values above the range investigated here, and for these individuals the model does predict weaker inbreeding depression in lambs compared to young adults (i.e. comparing $F_{roh} = 0.3$ with 0.4). Arguably, the most inbred individuals are also the part of the population where this would matter most, or at least it matters for these individuals, too. The conclusions drawn by the reviewer are therefore only correct for the range of F_{roh} values investigated ($F_{roh} < 0.3$ or 0.35). Overall, we would conclude that neither interpretation (inbreeding depression weaker or stronger) is predicted, but both directions are predicted for a certain range of the data. The reviewer might argue here that the larger F_{roh} range (say, $0.3-0.4$) is irrelevant, because only the first few age classes contain individuals falling into this range. However, we would argue here that in fact most datapoints overall ($>60\%$) are individuals of age 0-2, which is why these age classes are particularly relevant biologically and for the model estimation. In the end, this is also why the model estimates weaker inbreeding depression on the logit and odds-ratio scales.

For a more nuanced interpretation of these results, we have now re-written the relevant sections in the manuscript to completely interpret the $F_{roh} \times \text{lamb}$ interaction on the probability scale (lines 218 ff. and lines 386 ff.). We now conclude that the interpretation differs across the F_{roh} range, and that the effects of inbreeding depression in lambs are indeed subtle, with a need for future, more in-depth studies. We also removed the relevant sentence from the abstract.

Also, contrary to the claims in the text, the model does not give strong indications for a decline of inbreeding depression in adults, except perhaps in the very oldest age classes (e.g. age=7 above shows only an 8% decline in survival, compared to 11 and 13% in 4- and 1-year-olds). Since being a lamb is part of both age-related covariates, LAMB and AGE, it is impossible to say whether there is evidence for declining inbreeding depression in adults. The apparent decline could be exclusively due to the decline of inbreeding depression from lambs to adulthood.

R4: We believe that this is not correct. The model specifically tests whether inbreeding depression in lambs ($froh \times \text{lamb}$) is deviating from the estimated trendline of inbreeding depression across life ($froh \times \text{age}$), i.e. from the age predicted effect of F_{roh} .

As reported in the paper, the model does actually estimate weaker inbreeding depression in lambs on the logit scale (see also Supplementary Figure 8B), which is the opposite direction to what the reviewer suggests (because the reviewer's interpretation is based only on a subset of the full F_{roh} range on the probability scale, rather than the full range logit scale estimate which is what the model works with).

However, let's assume that indeed the only decline in inbreeding depression comes from the fact that lambs experience stronger inbreeding depression than all other age classes (and all other age classes would experience the same inbreeding depression). In this case, the $F_{roh} \times \text{lamb}$ predictor would explain all the variation and the $F_{roh} \times \text{age}$ predictor would not show any effect. However, not only does the $F_{roh} \times \text{lamb}$ interaction have a positive effect (and hence a trend in the opposite direction), but the $F_{roh} \times \text{age}$ predictor also shows a clear positive effect too, with credible intervals not overlapping zero. We are therefore confident that the model estimates and our interpretation with regards to the $F_{roh} \times \text{age}$ interaction are appropriate.

In summary, the model results as given in Supplementary Table 4 do not support the conclusions drawn in the main text. This discrepancy stems from the fact that the authors base much of their

interpretation in the text on the log-odds scale rather than the biologically meaningful survival scale. However, as I will discuss next, there are good reasons to think that the model used to estimate the parameters in Supplementary Table 4 does not fit the data well and that the parameter estimates therefore are off.

2) Supplementary Figure 9 shows the age-specific slope of FROH (β_{FROH}) on the log-odds scale when estimated in separate models for each age class. These estimates from separate models for each age class can be compared to the same slopes predicted from the model used in the manuscript. The R code at the end calculates these slopes from the parameter estimates given in Supplementary Table 4 and plots them for each age class. A comparison of this plot with Suppl. Figure 9 suggests that the model implemented in the manuscript does not fit the data well. The model implemented in the manuscript clearly overestimates the difference of β_{FROH} between lambs and one-year-olds, and Suppl. Figure 9 does not show the linear increase in β_{FROH} from age 1 to age 9 predicted by the model in the manuscript. All in all, this suggests that the model as implemented in the manuscript does not fit the data well and that the resulting parameter estimates are likely off.

R5: We regret having included Supplementary Figure 9. First of all, we do not think that the estimate from our full model should be validated against the estimates from many models which are less good. The main reason here is that estimating inbreeding depression for each age class does probably not work well, because the models for increasing ages are increasingly underpowered. Binary response variables such as survival do not contain much information, and therefore large sample sizes are needed to estimate complex models well. Also, proportionally (and absolutely) fewer individuals die with increasing age, leading to increasingly unbalanced groups and making it harder for the model to estimate the underlying effects. We have therefore excluded Figure 9 from the Supplementary as we now think that it is misleading. We apologize that the reviewer has spent time on this exercise.

Nevertheless, we take the reviewer's concerns about the fit of the model seriously, and have now made additional efforts to show that it does indeed fit well. In general, it is not entirely straightforward to diagnose the fit of a logistic regression model because many of the usual diagnostics with respect to residuals etc. do not work well. However, we are now showing that our model fits well to the data with a series of approaches, which we detail below. We also provide an RMarkdown script and pdf file with additional details for these methods: https://github.com/mastoffel/sheep_ID/blob/master/survival_model_diagnostics.pdf.

1) Posterior predictive checks

The idea here is to simulate new response vectors based on the fitted model and compare these to the empirical response (Andrew Gelman, Meng, & Stern, 1996). If the model fits well, it should produce survival data which is similar to our empirical data. To test this, we simulated survival data based on the fitted model, repeated this 1000 times and compared the simulated distribution to our empirical survival data. The ratio of estimated 1s and 0s (survival to deaths) in the simulated data is very close to what we see empirically - a good indication that the model fits well.

2) Binned residuals vs. fitted values

A usual model diagnostic is to plot residuals against fitted values, which is not very useful for logistic regression due to the discrete nature of the data points. An alternative is to divide the data into bins based on the fitted values and then plot the average residual versus the average fitted value for each bin. If the model fits well, we expect about 95% of the binned residuals to fall into the error bounds (A. Gelman & Hill, 2007, p.97). While the result varies slightly depending on the number of bins, around 93-96% of our bins fall within the error bounds for our data, again indicating a good fit to the data.

3) Predicting survival in new observations

Next, we wanted to know how well the model predicts the survival of new observations. To test this, we randomly chose 80% of the dataset as training data to fit the model. Based on the fitted model, we then predicted survival in the remaining 20% of the dataset. We used a simple measure of prediction accuracy, by calculating the proportion of correctly predicted 0s and 1s (deaths and survivals). We repeated this 100 times (see figure below). Overall, the model correctly predicts the death or survival of an animal in ~81% of cases, which we think is very high for predicting annual survival in a wild animal population.

Overall, we are therefore confident that our model fits well. While this does not imply that other models would not also fit well, we prefer the model in this manuscript, as it specifically addresses the hypotheses we were interested in.

Point 1) can be fixed by using the survival probability scale for the interpretation of the model output. It is more difficult for me to suggest how to fix point 2) given my ignorance of the raw data. However, given Supplementary Figure 9 I would suggest that a model with categorical age effects and the corresponding interactions with FROH might fit the data much better. This specification of the age effects makes the model a bit more parameter-rich, but given the large sample sizes the parameters should be well estimated.

R6: A categorical age effect would test inbreeding depression at every age only against inbreeding depression at a given reference level (possibly age = 0). This would result in 14 different tests only for comparing every age class to the reference level, which does not include other relevant comparisons, such as inbreeding depression at age 5 against age 2. Overall, the model would therefore require a large number of post hoc tests to explore how inbreeding depression changes across life.

Detailed comments:

- lines 204-206: the odds ratio of 0.32 does not represent the average effect on St. Kilda of a 10% increase in FROH. Instead, 0.32 is the odds ratio of a 0.1 increase in FROH in female, non-twins of age 3.5 years. It is not the average inbreeding depression experienced in the population. This should be made clear.

R7: We are now more specific and say that this is the estimate for an average aged , i.e. 2.4 year old non-twin male.

- lines 349-354: The odds ratio from this study is not comparable to estimates from other studies that are not based on odds ratios. In addition, as shown in point 1) above, odds ratios can be misleading on the survival scale: Note the poor correlation between odds ratios and inbreeding effects on the survival scale in this data set (R code below). It would be better to base this comparison on lethal equivalents.

R8: Between study comparisons are difficult based on either odds ratios or lethal equivalents, not only because the statistical methods are different but also because the inbreeding measures derived from microsatellites, pedigrees or SNPs as well as the fitness measures vary. We were careful in saying that genomic studies of species report stronger inbreeding depression compared to pre-genomic times while we also draw a very careful

comparison using lethal equivalents in the same paragraph. We therefore feel that we provide the reader with enough information to make up their own minds, while also making it clear that comparisons across studies and species are indeed difficult.

R Code:

=====

```
library(boot)
```

```
# average age according to Suppl. Figure 9
```

```
av_age <- (2134*1 + 1589*2 + 1282*3 + 1100*4 + 910*5 + 720*6 + 554*7 + 422*8 + 254*9)/(2134 + 1589 + 1282 + 1100 + 910 + 720 + 554 + 422 + 254)
```

```
# comment from authors: age class 0 (5591 individuals) is missing in the denominator.
```

```
# parameter estimates in linear predictor according to Suppl. Table 4 for males (sex=1) and not twins (twin=0)
```

```
int <- 3.34
```

```
Froh <- -1.14
```

```
Age <- -0.21
```

```
Lamb <- -3.41
```

```
Sex <- -0.63
```

```
fa_int <- 0.17
```

```
fl_int <- 0.62
```

```
# calculate linear predictor based on the parameter estimates above
```

```
# linear predictor for lamb (age=0) at average F_ROH (=0)
```

```
linpred_L_0 <- int + Froh*0 + Age*(0-av_age) + Lamb + Sex*1 + fa_int*(0-av_age)*0 + fl_int*0*1
```

```
surv_L_0 <- inv.logit(linpred_L_0)
```

```
surv_L_0
```

```
# linear predictor for lamb (age=0) at F_ROH of 0.34 (=1)
```

```
linpred_L_1 <- int + Froh*1 + Age*(0-av_age) + Lamb + Sex*1 + fa_int*(0-av_age)*1 + fl_int*1*1
```

```
surv_L_1 <- inv.logit(linpred_L_1)
```

```
surv_L_1
```

```
# inbreeding depression lambs
```

```
ID_Lamb <- (surv_L_0 - surv_L_1)/surv_L_0
```

```
# odds ratio for inbreeding effect in lambs
```

```
OR_Lamb <- (surv_L_1/(1-surv_L_1))/(surv_L_0/(1-surv_L_0))
```

```
# linear predictor for age=1 at average F_ROH (=0)
```

```
linpred_1_0 <- int + Froh*0 + Age*(1-av_age) + Sex*1 + fa_int*(1-av_age)*0
```

```
surv_1_0 <- inv.logit(linpred_1_0)
```

```
surv_1_0
```

```
# linear predictor for age=1 at F_ROH of 0.34 (=1)
```

```
linpred_1_1 <- int + Froh*1 + Age*(1-av_age) + Sex*1 + fa_int*(1-av_age)*1
```

```
surv_1_1 <- inv.logit(linpred_1_1)
```

```

surv_1_1

# inbreeding depression age=1
ID_1 <- (surv_1_0 - surv_1_1)/surv_1_0
# odds ratio for inbreeding effect in sheep age=1
OR_1 <- (surv_1_1/(1-surv_1_1))/(surv_1_0/(1-surv_1_0)))

# linear predictor for age=4 at average F_ROH (=0)
linpred_4_0 <- int + Froh*0 + Age*(4-av_age) + Sex*1 + fa_int*(4-av_age)*0
surv_4_0 <- inv.logit(linpred_4_0)
surv_4_0

# linear predictor for age=4 at F_ROH of 0.34 (=1)
linpred_4_1 <- int + Froh*1 + Age*(4-av_age) + Sex*1 + fa_int*(4-av_age)*1
surv_4_1 <- inv.logit(linpred_4_1)
surv_4_1

# inbreeding depression age=4
ID_4 <- (surv_4_0 - surv_4_1)/surv_4_0
# odds ratio for inbreeding effect in sheep age=4
OR_4 <- (surv_4_1/(1-surv_4_1))/(surv_4_0/(1-surv_4_0)))

# linear predictor for age=7 at average F_ROH (=0)
linpred_7_0 <- int + Froh*0 + Age*(7-av_age) + Sex*1 + fa_int*(7-av_age)*0
linpred_7_0
surv_7_0 <- inv.logit(linpred_7_0)
surv_7_0

# linear predictor for age=7 at F_ROH of 0.34 (=1)
linpred_7_1 <- int + Froh*1 + Age*(7-av_age) + Sex*1 + fa_int*(7-av_age)*1
linpred_7_1
surv_7_1 <- inv.logit(linpred_7_1)
surv_7_1

# inbreeding depression age=7
ID_7 <- (surv_7_0 - surv_7_1)/surv_7_0
# odds ratio for inbreeding effect in sheep age=2
OR_7 <- (surv_7_1/(1-surv_7_1))/(surv_7_0/(1-surv_7_0)))

# compare inbreeding depression and odds ratios
ID_Lamb
OR_Lamb
ID_1
OR_1
ID_4
OR_4
ID_7
OR_7

# Estimate age-specific beta_FROH (as in Suppl. Fig. 9) and plot them

```

```
beta_FROH <- rep(NA,10)
```

```
beta_FROH[1] <- Froh + fl_int + fa_int*(0-av_age)
```

```
beta_FROH[2] <- Froh + fa_int*(1-av_age)
```

```
beta_FROH[3] <- Froh + fa_int*(2-av_age)
```

```
beta_FROH[4] <- Froh + fa_int*(3-av_age)
```

```
beta_FROH[5] <- Froh + fa_int*(4-av_age)
```

```
beta_FROH[6] <- Froh + fa_int*(5-av_age)
```

```
beta_FROH[7] <- Froh + fa_int*(6-av_age)
```

```
beta_FROH[8] <- Froh + fa_int*(7-av_age)
```

```
beta_FROH[9] <- Froh + fa_int*(8-av_age)
```

```
beta_FROH[10] <- Froh + fa_int*(9-av_age)
```

```
plot(0:9,beta_FROH,xlab='age')
```

Reviewer #2 (Remarks to the Author):

Evaluation

I have read the revised manuscript from Stoffel et al. and find that this updated version is actually weaker than the original. Unfortunately, changes to the original manuscript have introduced a number of serious problems (see #1, 2, 5 below). Mostly, the manuscript is weakened by the attempt to link patterns of ROH density to selection. Upon deeper reflection, a couple of issues that were present in the original version have also occurred to me and now seem like bigger liabilities in light of the other changes (see #3, 4 below). My original assessment of the manuscript was that it represented a generally well-executed study of an important problem using an impressive dataset, constituting a genuine advance for the field. I still feel that the study is important, but the revised manuscript contains critical flaws that must be corrected.

R9: We thank the reviewer for another set of detailed comments on our revised manuscript, and are sorry to hear they find it weaker than the first draft, although it now incorporates the suggestions of the first round of reviews. After reading the comments carefully, we feel that there are some misunderstandings and hope to clarify these through our rebuttal and another revised version of the manuscript.

Major Issues

1. Unsupported claims about the role of selection in shaping patterns of ROH

Throughout the manuscript (lines 31, 67-72, 166-168, 326-346), the authors state that ROH density could be driven by selection, and they claim that their results support this hypothesis. Other hypotheses, apart from varying recombination rate, are not mentioned. I was especially dismayed to see that the abstract was rewritten to remove relevant details about the study, and a statement attributing patterns of ROH density to selection was added. The specific claim here is that there is selection either for or against ROH in certain parts of the genome, leading to regions with particularly high or low ROH density. In fact, there is no formal assessment of whether the ROH density could be generated under neutrality.

R10: We agree with the reviewer that we do not show that there is selection acting on ROH islands and deserts. However, we did not state this anywhere in the manuscript either. We

were careful to make clear that our analyses simply show that recombination is not the main driver creating ROH islands or deserts which qualifies these deserts and islands as *candidate regions* for natural selections (lines 332-333). In the abstract, we say that islands and deserts are "possibly" due to selection, which is a weak probabilistic word not usually perceived as high likelihood (see e.g. <https://hbr.org/2018/07/if-you-say-something-is-likely-how-likely-do-people-think-it-is>)

Furthermore, the reviewer says that "Other hypotheses, apart from varying recombination rate, are not mentioned". This is not correct. In lines 66-68 in the introduction, we explicitly state the known reasons for variation in ROH frequency across the genome (recombination, gene density, demographic stochasticity, linkage disequilibrium and selection).

Overall, we therefore feel that there has been a misunderstanding here. We appreciate that we have to communicate this more clearly and have now revised and clarified the interpretation of the potential causes of ROH density variation and in particular islands and deserts throughout the manuscript. First, of all, we have removed our statement that ROH deserts and islands are possibly due to selection from the abstract. We have rewritten the results section on ROH density and recombination rate to shift the focus to a better explanation on why this analysis is important and was suggested by reviewer 3 in the first place. Lastly, we also have rewritten the discussion and now specifically state that future studies are needed to evaluate selective explanations for ROH islands and deserts by comparison to simulated neutral scenarios. We hope that these changes put the interpretation of our findings in an unambiguous and clear framework.

The authors acknowledge that local recombination rate could affect power to detect long ROH, thus impacting ROH density estimation. However, varying recombination rate is not the only confounding factor, and it may not even be the most important factor (see more on the recombination rate analysis in point 2 below).

In the absence of selection for/against ROH, the density (probability) of ROH should partially reflect variable N_e across the genome, which is generated through the interaction of demographic history (drift, inbreeding, admixture), mutation rate, prior history of selection, and recombination rate. This is assuming technical artifacts are not artificially inflating diversity in some regions. Perhaps some regions really do have more or fewer ROH than expected under neutrality, but this must be tested explicitly before making claims about selection. The proper null model must control for the inherent variability in ROH likelihood across the genome. Specifically, highly variable regions of the genome would be less likely to appear in ROH, not because the probability of IBD is different here, but because the probability of inheriting two copies of the same haplotype by chance (IBS) is lower than in other regions with low diversity. It is not sufficient to simply quantify ROH density and then presume that the variability is driven by selection on ROH. There will always be a top 0.5% and a bottom 0.5% of ROH density. The question is whether these are the products of selection, and that has not been shown. Maybe the observed distribution is exactly what you would expect, just by the demographic process and whatever amount of variation there was to start.

At the very least, the authors should look at levels of diversity within versus between individuals (i.e. heterozygosity versus π). If within-individual diversity is lower/higher than expected, conditional on between-individual diversity and the expected homozygosity due to inbreeding, then this could point to selection. The idea is that detecting selection requires controlling for both inbreeding and random sampling. The claims about selection governing ROH density must be removed or qualified, or there needs to be a better analysis to provide evidence of selection.

Please see R10

2. Questionable interpretation of recombination rate analysis

The purpose of the recombination rate analysis (lines 166-188) is unclear. The first sentence of this section seems to imply that the purpose might be to control for recombination rate in order to strengthen the claim that ROH density outliers are driven by selection. However, maybe this analysis is just meant to reassure us that there is sufficient power to detect ROH, and recombination rate itself is not the primary driver of ROH density (although, as the authors note, high recombination rate itself is not expected to affect the true ROH density).

I don't think there's a real issue of power here. Power to detect ROH is mostly a problem when trying to identify short ROH (tens to hundreds of KB), not long ROH as in this study. But showing that there's only a mild effect of recombination rate on ROH density is good for covering the bases, and the results are consistent with what has been seen before (e.g. Pemberton et al. 2012, Figure 8B,C), despite this study using physical rather than genetic lengths. To summarize the weak correlation of recombination rate with ROH density, a brief mention in the text and a supplemental figure or two (as in the original manuscript) would suffice.

The introductory and closing sentences of this paragraph, however, and lines 331-333 of the discussion, make me question whether there was a different intended message. The paragraph ends with "ROH density mostly reflects the underlying patterns homozygosity along the genome." Isn't this just saying runs of homozygosity reflect homozygosity? Seems obvious that regions with high heterozygosity by definition must not have a lot of ROH. All else being equal, we would not expect recombination rate variation to drive ROH density (of long ROH), only the expected lengths of ROH, and even then, it would probably have little effect on ROH caused by recent inbreeding anyway. I can imagine a scenario where regions of high recombination rate might also harbor elevated diversity, which could in turn influence ROH prevalence, but this would need to be shown. I recommend reducing the recombination rate analysis to a brief note, or at least reframing to remove the implication that it shows evidence for selection on ROH.

R11: The analysis on the relationship between ROH density and recombination was a response to reviewer 3 (Marty Kardos), who rightly pointed out that "the relationship between ROH density and recombination rate is substantially more complicated than it appears by reading the relevant literature". At the core of this is the observation that extreme ROH densities could be found under varying recombination rates, even when the underlying IBD (heterozygosity) is constant (Figure 7 in Kardos, Qvarnström, & Ellegren, 2017). Consequently, in the most extreme case, one could imagine that the detected ROH islands and desert are purely a consequence of varying recombination rates, while their underlying IBD levels are average. The reason for this is that recombination by itself can not affect the true underlying IBD, while it will affect the density of detected ROH. This is also not a power issue, but rather an inherent property of all ROH analyses, as ROH are defined using a minimum length threshold. As this is an important issue, reviewer 3 has previously asked for a more in-depth analysis and a figure in the main paper. We agree with this and consequently decided to keep both the written paragraph and the figure in the main manuscript. However, we have now re-written the sections on the relationship between ROH density and recombination in the results (lines 166 ff.) and the discussion (lines 330 ff.) to further clarify the purpose of this analysis.

3. Lack of detail about demographic history in the main text

The original manuscript largely focused on the relationship between Froh and inbreeding depression. Despite the lack of detail about the system in the introduction of that version, I felt that the results were pretty straightforward, and the details were there in the discussion and methods. In the revised manuscript, there is a lot more attention on the mechanisms underlying patterns of ROH, and the GWAS results/interpretations are a little bit different. In light of these changes, it has become necessary to provide the history of the study system in detail in the introduction. Our expectations about patterns of ROH and the effect sizes of deleterious mutations (see point 4 also) are impacted by the fact that this population has undergone multiple bottlenecks, experienced admixture, and has unusually high levels of recent inbreeding. Also, the authors state that Soay sheep have been around for "thousands" of years. This is too vague. It is important to give proper context by disclosing these relevant details in the introduction. Lines 107-111 and 417-423 could simply be moved.

R12: We have now added a new paragraph on the history of the Soay sheep to the introduction and give a best guess of 4000 years ago for the arrival of Soay sheep on St. Kilda (Clutton-Brock & Pemberton, 2004).

4. Conclusions about effect sizes and number of lethal equivalents fail to properly take demographic history into account

The authors finds that inbreeding depression in this system is caused by deleterious alleles across many loci, with only a handful of mutations with large effect sizes. Further, the authors state that the number of lethal equivalents in Soay sheep is not particularly high, just moderate, and that perhaps this is due to better estimation from genomic data (lines 358-363). The major issue with both of these conclusions (small effect sizes, few lethal equivalents) is the failure to discuss the possibility that strongly deleterious mutations have been purged through repeated bottlenecks and long-term small N_e .

This population has experienced very high levels of inbreeding, multiple bottlenecks (relocation to St. Kilda, colonization of Soay, initial domestication of sheep), and admixture. All of these factors likely shaped the load of segregating deleterious mutations. It seems reasonable that with the intense inbreeding and drift, most of the remaining recessive deleterious alleles would be relatively weak, as large effect mutations would have already been eliminated by selection, or were preferentially lost through repeated bottlenecks that would be less likely to preserve low frequency alleles. It therefore makes sense that few loci of large effect were discovered here, and that the number of lethal equivalents is not especially high. This needs to be discussed in the text, in part to ground the results, but also to highlight that this is a unique system with a particular history, and the findings are probably not universal.

R13: We extensively discuss the expected relationship between small population size, admixture, purging and deleterious mutation in the last two paragraphs with respect to the GWAS results and the effect sizes of loci underpinning inbreeding depression. We do not explicitly discuss the number of lethal equivalents with respect to the Soay sheep demography mainly because we currently lack the data to appropriately compare this estimate to other species. As discussed in the text, studies differ widely in genetic and statistical methods and fitness measures, which is why we very carefully framed our statement on inbreeding load (lines 363 ff.). In particular, we do not want to speculate on the relationship between demographic events such as bottlenecks, purging and inbreeding

depression, as we do not explicitly test this and because this is a strongly debated topic at the moment (Ralls, Sunnucks, Lacy, & Frankham, 2020). As such, we feel that our discussion as it stands is appropriate and includes all important considerations with respect to of Soay sheep demography, admixture and the expectation of purging in a small population.

5. Data embargo is unacceptable

The new data availability statement indicates the data will be under embargo for one year, with no justification. For a study of potentially high significance that will be published in a widely-read and open access journal, the data must be made available with the paper. I cannot think of a reason why this dataset should be kept out of the public domain. This embargo is completely at odds with the spirit of an open access journal.

R14: We have now removed the one year embargo and will make the dataset available immediately.

Reviewer #3 (Remarks to the Author):

I am impressed by the revision and have only a few remaining minor comments below.

R15: Thank you!

138: insert 'more than' before 'twice'.

R16: Done.

142 (and other relevant text in the MS): I believe this actually refers to the largest fraction of the IBD part of the genome, not the largest fraction of ROH. It is not expected that ROH this long would outnumber shorter ones, as the distribution of ROH lengths is always expected to be exponential (i.e., many, many very short ROH and fewer long ROH), even if most of the genome is covered in very long ROH.

R17: Yes, we changed this throughout the MS.

Figure 4: You might consider putting spaces in between each chromosome, and making the points smaller to increase the clarity of the P-value distribution across each chromosome. The P-values on the right hand side of each chromosome are covered up by those on the left hand side of the following chromosome.

R18: We made the points slightly smaller and added spaces between the chromosomes, thanks for the suggestion.

Marty Kardos

References

- Clutton-Brock, T. H., & Pemberton, J. M. (2004). *Soay sheep: Dynamics and selection in an island population*. Cambridge University Press.
- Gelman, A., & Hill, J. (2007). *Data analysis using regression and multilevel/hierarchical models*. Cambridge, U.K.: Cambridge University Press.
- Gelman, Andrew, Meng, X.-L., & Stern, H. (1996). Posterior predictive assessment of model fitness via realized discrepancies. *Statistica Sinica*, 733–760.
- Kardos, M., Qvarnström, A., & Ellegren, H. (2017). Inferring individual inbreeding and demographic history from segments of identity by descent in *Ficedula* flycatcher genome sequences. *Genetics*, 205(3), 1319–1334.
- Ralls, K., Sunnucks, P., Lacy, R. C., & Frankham, R. (2020). Genetic rescue: A critique of the evidence supports maximizing genetic diversity rather than minimizing the introduction of putatively harmful genetic variation. *Biological Conservation*, 251, 108784. doi: 10.1016/j.biocon.2020.108784

REVIEWER COMMENTS

Reviewer #1 (Remarks to the Author):

Genetic architecture and lifetime dynamics of inbreeding depression in a wild mammal

Two statistical issues were unresolved in the last version of the manuscript: 1) the interpretation of the model results in the main text, and 2) the poor fit of the age-specific effects of the model when compared to Supplementary Figure 9 of the last submission.

The authors have changed the interpretation in the text slightly, but the interpretation of inbreeding depression as odds-ratios is still prominent (e.g. lines 213-219). More importantly, the statistical model remains unchanged. As a consequence, serious problems remain.

I have explained these problems in detail in my last review and there is no need to repeat them here. Instead, I will directly address some of the issues that arise from the replies of the authors.

R3: The pattern seen here is not based on evidence in the data but is instead a reflection of a particular assumption of the model. The authors deduce from the patterns in Figure 3C that highly inbred lambs exhibited weaker inbreeding depression than less inbred lambs. However, the model (lines 537-538) forces inbreeding depression to be constant across all levels of inbreeding, as can be seen from the specification of inbreeding depression as linear effects in the model. Such a constant effect on the logit scale results in the non-linear patterns on the probability scale shown in Figure 3. Thus, the fact that inbreeding depression appears to be weaker in more inbred lambs is not a pattern in the data but a direct reflection of the modeling choices. In other words, what the authors interpret here as a biological pattern is in fact simply a reflection of an untested model assumption. Evidence for differences in inbreeding depression at different levels of inbreeding can only be obtained with a model that allows inbreeding depression to vary as a function of inbreeding level.

R4: I could not really follow the reply here. The main issue remains, namely that FROH*LAMB and FROH*AGE are confounded, because being a lamb is part of both AGE and LAMB. As a consequence, the linear trend of inbreeding depression across life is itself a function of the effects of being a lamb, and the two cannot be separated. Without removing this confounding, I cannot see how one can be confident about the model results and interpretations.

R5: I think the reply here does not address the crucial point. The issue at hand is not statistical power but parameter estimation. The previous supplementary figure 9 clearly showed that models run separately for each age class yield very different age-specific patterns of inbreeding depression than the full model presented in the manuscript. This can be seen by comparing the previous supplementary figure 9 to the age effects estimated by the full model, which I plotted with the R code in my last review.

Thank you for pointing out that I had omitted the lambs in the denominator. I have fixed this, but the age-dependent estimates of inbreeding depression still look completely different from those previously reported in supplementary figure 9. This has nothing to do with low statistical power, and I cannot see how separate models would yield less good parameter estimates. Instead, this comparison shows that the full model does not do a good job parametrising the age-specific inbreeding effects. Removing supplementary figure 9 from the manuscript does not solve the problem, it only hides it.

I do appreciate the efforts the authors have put into assessing model fit. However, the focus of these assessment is not on the age-specific effects and, thus, they do not show whether or not the model fits well. Checks were carried out for the overall proportion of survival (1), the distribution of the residuals (2), and prediction accuracy averaged across the entire data set (3). None of these tests ask whether the age-dependent effects of inbreeding are correctly predicted, which is what this manuscript focuses on and what seems to be poorly predicted.

R6: Again, the authors focus here on hypothesis testing rather than on effect size estimation. It's the latter that is biologically meaningful. In addition, in contrast to the claim here the current model does not allow a statistical comparison of inbreeding depression at age 5 against age 2.

Reviewer #3 (Remarks to the Author):

I believe the authors' revisions in response to reviewer comments have improved the manuscript considerably, and I suggest accepting the paper for publication. In particular, the material added with respect to the relationship between ROH density and recombination advances understanding of mechanisms leading to genome-wide variation in the abundance of IBD segments, and thus adds to the importance of the manuscript. Regarding a previous concern raised by reviewer 2, I believe that the discussion of the potential for selection to affect this distribution is balanced well against the alternative hypotheses in the current revision. I do have one suggestion that I hope will help to improve the discussion of the results on the genome-wide distribution of ROH:

It would be helpful to discuss (eg in the paragraph beginning on line 330) why you think there is a weak relationship between ROH abundance and recombination rate. In particular, it appears (as mentioned previously by another reviewer), that this is likely because this analysis is restricted to very long ROH (>1.2 Mb). Had the analysis included short ROH (<1Mb, as is possible with WGS data) it is likely that this relationship would have strengthened; Haplotypes comprising shorter ROH have longer coalescent times on average, and are therefore more subject to having their lengths affected recombination events before coalescence. Another likely explanation is biological rather than technical: Soays are quite highly inbred, which means that the distribution of coalescent times across the genome must be truly shifted to the left compared to other species with a stronger r -ROH density relationship. This means there is on average much less chance for recombination to affect ROH lengths in Soays compared to some other study populations.

Other specific comments:

65-68: you could replace 'demographic stochasticity' (which implies random changes in population size) with 'genetic drift' (which implies random changes in haplotype frequency), as it is indeed the latter that can cause some regions to have very high ROH density and other related phenomena (increased LD, shifted site frequency spectrum and other signals often interpreted as signatures of selection).

136: add 'length' after ROH.

326: Change 'non-inbred' to 'least inbred'. All of the sheep are inbred to some degree.

333: I would replace 'demographic stochasticity' with genetic drift. The key here is that strong genetic drift can cause very high variance in haplotype diversity across the genome which can then lead to a high variance in the abundance of ROH across the genome.

Marty Kardos

Other comments by Reviewer 3 in response to Reviewer 1:

The reviewer has raised issues that are worth addressing in my opinion. The issues raised relate to only one of the major sections of the manuscript, and the issues will be easy for the authors to address. The modelling of age-specific inbreeding depression is difficult and necessarily requires a rather complex model (lots of predictors), and the best way to parameterize such a model is not immediately obvious.

I suggest that the authors revise the MS to present the material in Figure 3 and in the text (lines 199-237) on the probability scale only. I agree with the other reviewer that it is this scale that is most informative biologically.

The reviewer also argues that the authors have misinterpreted the results as showing that inbreeding depression is weaker in lambs than in the older age classes. Here I believe the reviewer is incorrect. Supplementary Figure 8 and Figure 3B show this clearly. The total reduction in survival (on both the probability scale [panel A] and the log-odds scale [panel B] in Supp Fig 8) with increasing inbreeding (from its minimum value to its maximum value) is larger for age 1 individuals (approximately 60% by eye) than for lambs (age 0, approximately 50% by eye). This is not the case when only considering later age classes (e.g., perhaps ages 3 and up) in comparison to lambs (Supp Fig 8)... However, there are far fewer individuals in the later age classes (most sheep die young) so this is not a problem biologically or statistically, but it is probably worth the authors mentioning in the paper that the signal of weaker inbreeding depression in the lambs is driven mostly by the comparison of lambs versus one and two year olds.

Perhaps the simplest and best solution to dealing with the modelling issues related to age-specific

inbreeding depression is to focus on a simpler model where the number of age classes considered is reduced substantially (e.g., include only lambs, age 1, and age 2+ individuals as age classes, or even just lambs versus non lambs). It is clearly hard to estimate the age-specific inbreeding depression for the later age classes because there are fewer old individuals, and no highly inbred old individuals. For example, there are only five individuals age 3+ with Froh > 0.3 (Fig 3A). I believe my suggested approach would eliminate most of the problems raised by the other reviewer. Alternatively, the authors could repeat the cross validation exercise to see if the present model can reliably predict age-specific inbreeding depression. However, I doubt that this will work well at all considering the absence of highly inbred old individuals in the data. I suspect that the end result will be a model that includes the Froh x lamb term, and no Froh x age term.

The reviewer thought that the patterns in Figure 3C have likely arisen from the modelling decisions instead of real patterns in the data. I am always favor showing the data in the rawest possible form alongside model results. Perhaps an effective solution is therefore to first hold as many extraneous variables as possible constant (e.g., include only males and non-twins), and plot confidence intervals for the proportion of survivors in different bins of Froh for different age classes (eg the greatly reduced number of age classes as suggested above). This may be an effective way of summarizing the raw data outside of the (necessarily) pretty complex model presented in the manuscript.

The reviewer suggests that the collinearity of two predictors in the mixed model (Froh x age, and Froh x lamb) makes it difficult to reliably interpret the model results. These two predictors are clearly not independent so I agree. If the authors insist on including both of these terms in their analyses, a sensible solution may be to run the analysis twice: once including Froh x lamb, and once including Froh x age, and see if the major inferences change (I strongly doubt anything will change substantively). Additionally, if the authors can make a reasoned biological argument that we expect (a priori) different situations for lambs versus individuals age > 0, they could potentially model Froh x age as a predictor while excluding age 0 individuals.

The reviewer suggests that difference between the previous Supplementary Figure 9 and the results presented in the manuscript suggest that the current model fits the data poorly. I agree with the reviewer that the cross-validation exercise does not suggest that the age-specific effects are estimated well, and agree that the performance of the model in estimating the age-specific effects should be evaluated... Greatly reducing the considered age classes will improve the predictive power of the model considerably.

I would support publishing it in any high profile journal. I agree with the other reviewer that the modelling can be improved though.

I also think that the solutions are rather easy and well within reach for the authors. I suggest asking the authors to revise the manuscript according to my suggestions above. I am also happy to discuss the issues above with the other reviewer if everyone agrees that it would be helpful.

Again, we thank the reviewers for their comments. We have now made broader changes to the statistical modelling and manuscript which we believe address all remaining concerns. We first summarise the changes to the statistical model and manuscript based on the comments from Reviewers 1 & 3, followed by brief point-by-point responses to the reviewers suggestions.

Summary of the changes to the statistical model

To summarise, there are two main concerns about the statistical model: (1) That there is no decrease of inbreeding depression with increasing age (or that the effects merely result from a lamb/non-lamb comparison), and (2) That there is no relaxed inbreeding depression in lambs. In our revised model and manuscript, we now focus on point (1). Point (2) is indeed subtle (as we have previously acknowledged in the manuscript), and we have now removed that line of argumentation from the manuscript.

As suggested by reviewer 3, we now plotted the raw data as the proportion of survivors in different life stages and for different F_{ROH} classes. To get a meaningful and interpretable plot for illustration purposes with sufficient sample sizes per group, we clustered all sheep into four life stages, lamb (age = 0), early life (1,2), mid-life (3,4) and late life(5+). In addition, we clustered F_{ROH} into bins, in continuous steps of 0.02 from the minimum 0.18 to 0.28, which is the range in which most (95%) of the data fall. We clustered the remaining data into one, broader 'inbred' category (0.28-0.5). This allows a comparison between life stages, in a way in which each life stage still consists of a large enough number of individuals in the highest F_{ROH} category. The raw data shows that inbreeding depression decreases over time, with old individuals barely showing any difference even between the least and most inbred F_{ROH} classes in terms of the proportion of survivors (new Figure 3B, see below). In addition, to get a feeling for the sampling variance, we used 10 iterations of non-parametric bootstrapping. These are plotted as transparent points in Figure 3B, and are largely overlapping with the point-estimates, reflecting very little sampling variance due to large sample sizes in each group.

As a next step, we modelled inbreeding depression in survival with the following new model (R formula notation):

$$\text{Annual survival} \sim F_{ROH} * \text{life_stage} + \text{sex} + \text{twin} + (1|ID) + (1|mum_id) + (1|year) + (1|pedigree)$$

The life_stage variable is a factor with 4 levels, lamb, early life, mid-life and late life, with "lamb" as the reference level. Consequently, the model estimates (a) three main effects for the difference in survival probability between lamb and each of the three life stages and (b) three interaction effects for the differences in the effects of F_{ROH} on survival (i.e. inbreeding depression) between the lamb stage and each of the three following life stages.

Both the predicted survival probabilities (Figure 3C) and the logit / odds-ratio estimates ($F_{ROH} * \text{life stage}$ interactions in the new Supplementary Table 4) show that there is a decrease in the strength of inbreeding depression in mid / late life. We believe that the pattern of decreasing inbreeding depression across life both in the raw data and through consistent model estimates based on two different modelling strategies is convincing evidence. We focus now on the above model and show the two raw data plots alongside the predicted survival probabilities in the main Figure 3, while we also show an extended summary table for the model estimates, including all estimates on the log-odds and odds-ratio scales in a new Supplementary Figure 4.

As a consequence, we have re-written the respective parts in the manuscript, including the results (lines 217 ff), methods (lines 524 ff) and the discussion, where we removed sections with respect to inbreeding depression in lambs, and focused on the decrease in the strength of inbreeding depression in mid/late life (lines 375 ff).

Figure 3: Inbreeding depression in annual survival. Panel A shows the distributions of inbreeding coefficients F_{ROH} in Soay sheep age classes ranging from 0 to 9 years. Panel B shows the proportion of surviving individuals per year in four different life stages and among different F_{ROH} classes. As highly inbred individuals are relatively rare, the last class spans a wider range of inbreeding coefficients. In addition to the point-estimates, 10 non-parametric bootstrap estimates across individuals are shown as transparent points, which are largely overlapping with the point estimate, reflecting large within-group sample sizes. Panel C shows the predicted survival probability and 95% CI over the range of inbreeding coefficients F_{ROH} for each life stage, while holding sex and twin constant at 1 (male) and 0 (no twin). The predictions for the later life stages classes exceed the range of the data (panel A) but are shown across the full range for comparability.

Term	Post.Mean	Std.Error	CI (2.5%)	CI (97.5%)	Standardisation	Info
Fixed effects						
Intercept	0.53 (1.7)	0.44 (1.55)	-0.32 (0.73)	1.42 (4.14)		
F_{ROH}	-0.91 (0.4)	0.14 (1.15)	-1.2 (0.3)	-0.63 (0.53)	(x * 10)-mean(x * 10)	continuous
LifeStage: EarlyLife	2.82 (16.78)	0.08 (1.08)	2.66 (14.3)	3 (20.09)		categorical (0=no, 1=yes)
LifeStage: MidLife	3.61 (36.97)	0.13 (1.14)	3.38 (29.37)	3.91 (49.9)		categorical (0=no, 1=yes)
LifeStage: LateLife	2.41 (11.13)	0.19 (1.21)	2.14 (8.5)	2.91 (18.36)		categorical (0=no, 1=yes)
Sex	-0.56 (0.57)	0.05 (1.05)	-0.66 (0.52)	-0.46 (0.63)		categorical (0=female, 1=male)
Twin	-0.74 (0.48)	0.07 (1.07)	-0.88 (0.41)	-0.61 (0.54)		categorical (0=no, 1=yes)
F_{ROH} * (LifeStage: EarlyLife)	-0.21 (0.81)	0.26 (1.3)	-0.72 (0.49)	0.3 (1.35)		
F_{ROH} * (LifeStage: MidLife)	0.12 (1.13)	0.38 (1.46)	-0.62 (0.54)	0.88 (2.41)		
F_{ROH} * (LifeStage: LateLife)	0.5 (1.65)	0.28 (1.32)	-0.05 (0.95)	1.05 (2.86)		
Random effects (variances)						
Birth year	0.79	0.18	0.59	1.24		n = 40
Capture year	2.57	0.17	2.36	2.97		n = 40
Individual	0.3	0.01	0.29	0.33		n = 5952
Add. genetic	0.3	0.02	0.28	0.34		Pedigree-based

Supplementary Table 4: Model estimates for the Bayesian binomial animal model of annual survival. Shown are the posterior mean, standard error, lower and upper credible interval on the logit (log-odds) scale with the exponentiated estimates (odds-ratios) in round brackets, and information about the standardisation of the variables. Life stage was fitted as a factor with four levels, lamb (age = 0, reference level), early life (age = 1,2), mid-life (age = 3,4), late life (5+). The last column shows whether

variables were fitted as continuous or categorical and how the levels for categorical variables were coded. For the random intercept effects of birth year, capture year and individual, the last column shows the number of groups. The last row in the table shows the estimates for the additive genetic variance, based on a pedigree-derived relationship matrix. The dataset underlying the model contained 15889 observations from 5952 individuals.

Full reviewer comments / Point-by-point responses

Reviewer #1 (Remarks to the Author):

Genetic architecture and lifetime dynamics of inbreeding depression in a wild mammal

Two statistical issues were unresolved in the last version of the manuscript: 1) the interpretation of the model results in the main text, and 2) the poor fit of the age-specific effects of the model when compared to Supplementary Figure 9 of the last submission.

The authors have changed the interpretation in the text slightly, but the interpretation of inbreeding depression as odds-ratios is still prominent (e.g. lines 213-219). More importantly, the statistical model remains unchanged. As a consequence, serious problems remain.

I have explained these problems in detail in my last review and there is no need to repeat them here. Instead, I will directly address some of the issues that arise from the replies of the authors.

R1: As described above, we have now changed our statistical model, which we think is more straightforward to interpret now, and close to what the reviewer has suggested initially (fitting age as a categorical variable). The model estimates on the probability and logit scales predict a decrease in the strength of inbreeding depression in mid-life and late life, consistent with a new plot of the raw data (Figure 3B) and the estimated $F_{ROH} * \text{age}$ interaction from the previous model. We now focus on this decrease in inbreeding depression in mid-life and late life and have removed our discussion on weaker inbreeding depression in lambs.

R3: The pattern seen here is not based on evidence in the data but is instead a reflection of a particular assumption of the model. The authors deduce from the patterns in Figure 3C that highly inbred lambs exhibited weaker inbreeding depression than less inbred lambs. However, the model (lines 537-538) forces inbreeding depression to be constant across all levels of inbreeding, as can be seen from the specification of inbreeding depression as linear effects in the model. Such a constant effect on the logit scale results in the non-linear patterns on the probability scale shown in Figure 3. Thus, the fact that inbreeding depression appears to be weaker in more inbred lambs is not a pattern in the data but a direct reflection of the modeling choices. In other words, what the authors interpret here as a biological pattern is in fact simply a reflection of an untested model assumption. Evidence for differences in inbreeding depression at different levels of inbreeding can only be obtained with a model that allows inbreeding depression to vary as a function of inbreeding level.

R2: We have now removed that line of argumentation from the manuscript, to instead focus on weaker inbreeding depression in later life stages.

R4: I could not really follow the reply here. The main issue remains, namely that $FROH * LAMB$ and $FROH * AGE$ are confounded, because being a lamb is part of both AGE and LAMB. As a consequence, the linear trend of inbreeding depression across life is itself a function of the effects of being a lamb, and the two cannot be separated. Without removing this confounding, I cannot see how one can be confident about the model results and interpretations.

R3: We believe that our new model is more straightforward to interpret, whilst showing the same trend across life.

R5: I think the reply here does not address the crucial point. The issue at hand is not statistical power but parameter estimation. The previous supplementary figure 9 clearly showed that models run separately for each age class yield very different age-specific patterns of inbreeding depression than the full model presented in the manuscript. This can be seen by comparing the previous supplementary figure 9 to the age effects estimated by the full model, which I plotted with the R code in my last review. Thank you for pointing out that I had omitted the lambs in the denominator. I have fixed this, but the age-dependent estimates of inbreeding depression still look completely different from those previously reported in supplementary figure 9. This has nothing to do with low statistical power, and I cannot see how separate models would yield less good parameter estimates. Instead, this comparison shows that the full model does not do a good job parametrising the age-specific inbreeding effects. Removing supplementary figure 9 from the manuscript does not solve the problem, it only hides it.

R4: Our new model predictions in Figure 3C and the new raw data plot in Figure 3B are addressing this concern in showing that inbreeding depression becomes weaker in later life. We do not intend to hide any information, which is why we provide the full analysis code and the raw data, and why we are showing plots of the raw data from two different perspectives (3A, B), the model predictions on the probability scale (3C) and the model estimates on the odds-ratio and logit scales (Supplementary Table 4).

I do appreciate the efforts the authors have put into assessing model fit. However, the focus of these assessment is not one the age-specific effects and, thus, they do not show whether or not the model fits well. Checks were carried out for the overall proportion of survival (1), the distribution of the residuals (2), and prediction accuracy averaged across the entire data set (3). None of these tests ask whether the age-dependent effects of inbreeding are correctly predicted, which is what this manuscript focuses on and what seems to be poorly predicted.

R6: Again, the authors focus here on hypothesis testing rather than on effect size estimation. It's the latter that is biologically meaningful. In addition, in contrast to the claim here the current model does not allow a statistical comparison of inbreeding depression at age 5 against age 2.

R5: Our focus in the manuscript is now on estimating the strength of inbreeding depression and discussing a decrease in the effects of inbreeding depression in mid/late-life. These patterns are clear in the raw data and predicted by both the old and new model, on both the logit and the probability scale. We therefore hope the reviewer agrees that our interpretation is acceptable now.

Reviewer #3 (Remarks to the Author):

I believe the authors' revisions in response to reviewer comments have improved the manuscript considerably, and I suggest accepting the paper for publication. In particular, the material added with respect to the relationship between ROH density and recombination advances understanding of mechanisms leading to genome-wide variation in the abundance of IBD segments, and thus adds to the importance of the manuscript. Regarding a previous concern raised by reviewer 2, I believe that the discussion of the potential for selection to affect this distribution is balanced well against the alternative hypotheses in the current revision. I do have one suggestion that I hope will help to improve the discussion of the results on the genome-wide distribution of ROH:

It would be helpful to discuss (eg in the paragraph beginning on line 330) why you think there is a weak relationship between ROH abundance and recombination rate. In particular, it appears (as mentioned previously by another reviewer), that this is likely because this analysis is restricted to very long ROH (>1.2 Mb). Had the analysis included short ROH (<1Mb, as is possible with WGS data) it is likely that this relationship

would have strengthened; Haplotypes comprising shorter ROH have longer coalescent times on average, and are therefore more subject to having their lengths affected recombination events before coalescence. Another likely explanation is biological rather than technical: Soays are quite highly inbred, which means that the distribution of coalescent times across the genome must be truly shifted to the left compared to other species with a stronger r-ROH density relationship. This means there is on average much less chance for recombination to affect ROH lengths in Soays compared to some other study populations.

We have now investigated this by re-running the analysis with two more ROH datasets: One with a minimum ROH length of 0.4Mb, and one with a minimum ROH length of 3Mb. Note that the SNP density in our dataset is sufficient to call ROH with a minimum length of 0.4Mb, which are still expected to contain around ~50 SNPs and could therefore be called with the same PLINK parameters as in the original analysis.

Recombination rate variation explained 4% of ROH density in the original dataset with a minimum ROH length of 1.2Mb. When also including shorter ROH, recombination only explains 1% of the variation in ROH density, while recombination explains 8% of the ROH abundance in the dataset containing only longer ROH. Consequently, recombination is affecting the detected density/abundance of longer ROH more strongly than short ROH. The reason for this might be the following: Long ROH are overall less abundant across the genome, while they are (because of their length) more likely to be broken down by recombination. Proportionally to their overall abundance, recombination might therefore have a larger impact on long ROH than on short ROH (which are much more common). In a way, the reason for this might therefore be both technical and biological. If the background relatedness/inbreeding levels were lower, this pattern might be different. We have added the analysis to the results (lines 189 ff), and extended the discussion (lines 335 ff).

Other specific comments:

65-68: you could replace 'demographic stochasticity' (which implies random changes in population size) with 'genetic drift' (which implies random changes in haplotype frequency), as it is indeed the latter that can cause some regions to have very high ROH density and other related phenomena (increased LD, shifted site frequency spectrum and other signals often interpreted as signatures of selection).

136: add 'length' after ROH.

326: Change 'non-inbred' to 'least inbred'. All of the sheep are inbred to some degree.

333: I would replace 'demographic stochasticity' with genetic drift. The key here is that strong genetic drift can cause very high variance in haplotype diversity across the genome which can then lead to a high variance in the abundance of ROH across the genome.

We have changed / addressed all of these as suggested, thanks for the attention to detail!

Marty Kardos

REVIEWERS' COMMENTS

Reviewer #1 (Remarks to the Author):

Thank you for the revisions.

The new statistical model simplifies interpretations and yields age-related patterns of inbreeding depression that are in line with the previous supplementary figure 9.

I am satisfied with the revision and have no additional comments.